# Phidias: A Generative Model for Creating 3D Content from Text, Image, and 3D Conditions with Reference-Augmented Diffusion

**Zhenwei Wang**[1][†][*] **Tengfei Wang**[2][‡][*] **Zexin He**[3][†] **Gerhard Hancke**[1] **Ziwei Liu**[4] **Rynson W.H. Lau**[1][‡]
[1]City University of Hong Kong   [2]Shanghai AI Lab   [3]CUHK   [4]S-Lab, NTU

## Abstract

Generative 3D modeling has made significant advances recently, but it remains constrained by its inherently ill-posed nature, leading to challenges in quality and controllability. Inspired by the real-world workflow that designers typically refer to existing 3D models when creating new ones, we propose *Phidias*, a novel generative model that uses diffusion for reference-augmented 3D generation. Given an image, our method leverages a retrieved or user-provided 3D reference model to guide the generation process, thereby enhancing the generation quality, generalization ability, and controllability. *Phidias* integrates three key components: 1) *meta-ControlNet* to dynamically modulate the conditioning strength, 2) *dynamic reference routing* to mitigate misalignment between the input image and 3D reference, and 3) *self-reference augmentations* to enable self-supervised training with a progressive curriculum. Collectively, these designs result in significant generative improvements over existing methods. *Phidias* forms a unified framework for 3D generation using text, image, and 3D conditions, offering versatile applications. Project page: https://RAG-3D.github.io/.

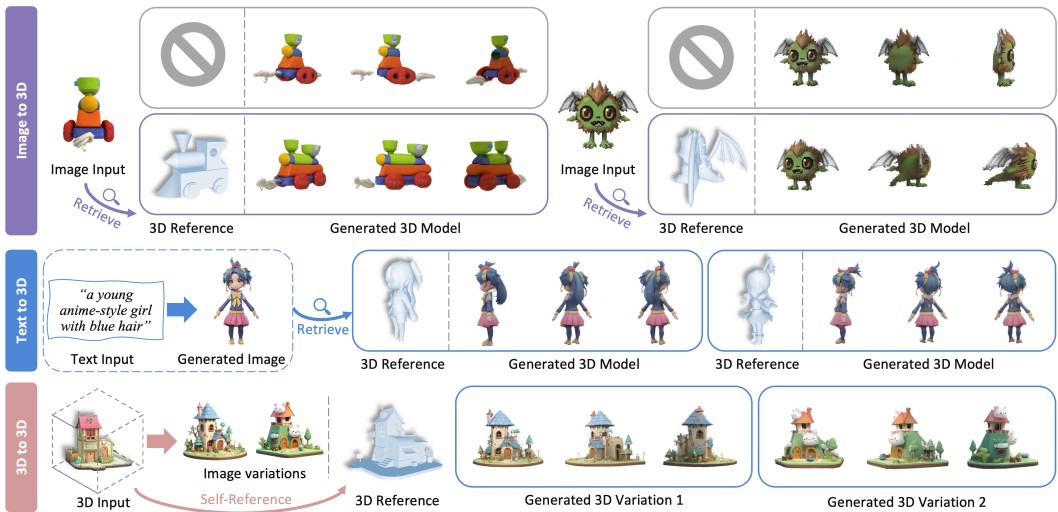

Figure 1: The proposed method, *Phidias*, can produce high-quality 3D assets given 3D references, which can be obtained via retrieval (top two rows) or specified by users (bottom row). It supports 3D generation from a single image, a text prompt, or an existing 3D model.

## 1 Introduction

The goal of 3D generative models is to empower artists and even beginners to effortlessly convert their design concepts into 3D models. Consider the input image in Fig. 1. A skilled craftsman can, through a blend of skills and creativity, convert a 2D concept image into an exquisite 3D model. This creative process may originate from artists' pure imagination or, more commonly, through examining one or more existing 3D models as a source of inspiration (Bob, 2022; Carvajal, 2023). Artists

---

[†]Intern at Shanghai AI Lab. [*]Equal Contribution. [‡]Corresponding Authors.

often reference pre-existing 3D models to help improve the modeling quality. The question then arises: could we develop a reference-based 3D generative model that can replicate this capability?

Over the years, a plethora of works (Wang et al., 2023; Liu et al., 2023b; Hong et al., 2023; Bensadoun et al., 2024) steadily expanded the frontiers of 3D generative models. These methods, while yielding stunning performance, still face several challenges. **1) Generation quality.** A single image cannot furnish sufficient information for reconstructing a full 3D model, due to the ambiguity of this ill-posed task. It necessitates the generative model to "hallucinate" the unseen parts in a data-driven manner. However, this hallucination step can lead to view inconsistency and imprecise geometries that appear abrupt and unrealistic. **2) Generalization ability.** These models often struggle with out-of-domain cases, such as atypical input views or objects, constrained by the data coverage of existing 3D datasets (Deitke et al., 2023). Meanwhile, the growing variety and quantity of object categories exacerbate the difficulty for generative models to learn implicit shape priors, with a limited model capacity v.s. an infinitely diverse array of objects. **3) Controllability.** Due to the ambiguity, one input image can produce several plausible 3D models, each differing in shape, geometric style, and local patterns. Existing methods are constrained by limited diversity and controllability, which hinders the ability to predictably generate the desired 3D models.

To address these challenges, we propose to take 3D models as additional inputs to guide the generation, inspired by the success of retrieval augmented generation (RAG) in language (Lewis et al., 2020) and image (Sheynin et al., 2022). Given an input image and a reference 3D model, we present *Phidias*, a novel reference-augmented diffusion model that unifies 3D generation from text, image, and 3D conditions. As shown in Fig. 1, the reference 3D model would help 1) **improve generation quality** by alleviating ambiguity with richer information for unseen views, 2) **enhance generalization capacity** by serving as a shape template or an external memory for generative models, and 3) **provide controllability** by allowing users to indicate desired shape patterns and geometric styles.

Our method introduces a reference-augmented multi-view diffusion model, followed by sparse-view 3D reconstruction. The goal is to produce 3D models faithful to the input concept image, with improved quality by incorporating relevant information from the 3D reference. However, it is nontrivial to learn such a generative model due to the *Misalignment Dilemma*, where the discrepancy between the concept image and the 3D reference can lead to conflicts in the generation process. This requires our model to utilize the misaligned 3D reference adaptively. To tackle this challenge, *Phidias* leverages three key designs outlined below.

The first is **meta-ControlNet**. Consider the 3D reference as conditions for the diffusion model. Unlike image-to-image translation works (Zhang et al., 2023; Wang et al., 2022) that demand the generated images to closely follow the conditions, we treat the reference model as auxiliary guidance to provide additional information. The generated multi-view images are expected to be consistent with the concept image, without requiring precise alignment with the reference model. To this end, we build our method on ControlNet and propose a meta-control network to dynamically modulate the conditioning strength when it conflicts with the concept image, based on their similarity.

The second design is **dynamic reference routing** for further alleviating the misalignment. Rather than using the same 3D reference for the full diffusion process, we adjust its resolution across denoise timesteps. This follows the dynamics of the reverse diffusion process (Balaji et al., 2022), which generates the coarse structure in high-noised timesteps and details in low-noised timesteps. Thus, we can alleviate the generation conflicts by starting with a coarse 3D reference and progressively increasing its resolution as the reverse diffusion process continues.

The final key design is **self-reference augmentations**. As it is not feasible to gather a large set of 3D models together with their matching references, a practical solution is to use the 3D model itself as its own reference (*i.e.,* self-reference) for self-supervised learning. The trained model, however, may not work well when the 3D reference does not align with the concept image. To alleviate this problem, we apply a variety of augmentations to the 3D models to simulate this misalignment. In addition, we also introduce a progressive augmentation approach that leverages curriculum learning for the diffusion model to effectively utilize 3D references with varying degrees of similarity.

Taken together, the above ingredients work in concert to enable *Phidias* to achieve stunning performances in 3D generation. Several application scenarios are thus supported: 1) Retrieval-augmented image-to-3D generation, 2) Retrieval-augmented text-to-3D generation, 3) Theme-aware 3D-to-3D generation, 4) Interactive 3D generation with coarse guidance, and 5) High-fidelity 3D completion.

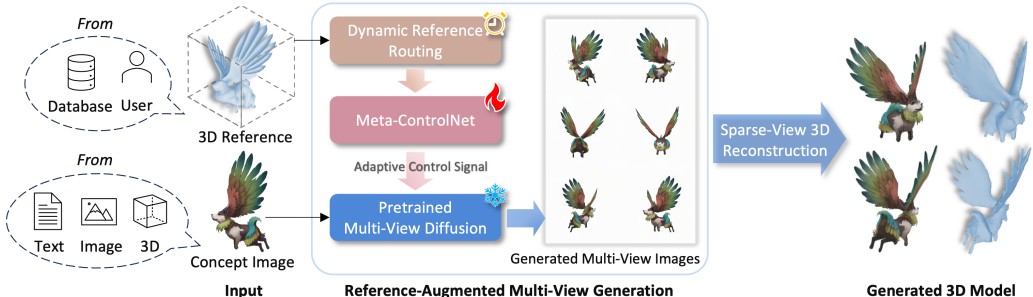

Figure 2: Overview of our *Phidias* model. It generates a 3D model in two stages: (1) reference-augmented multi-view generation and (2) sparse-view 3D reconstruction.

We summarize our contributions as follows: 1) We propose the first reference-based 3D-aware diffusion model. 2) We design our model with three key component designs to enhance the performance. 3) Our model serves as a unified framework for 3D generation, which enables a variety of applications with text, image, and 3D inputs. 4) Extensive experiments show that our method outperforms existing approaches qualitatively and quantitatively.

## 2 RELATED WORKS

**Image to 3D.** Pioneering works (Melas-Kyriazi et al., 2023; Tang et al., 2023; Chen et al., 2024b) perform 3D synthesis by distilling image diffusion priors (Poole et al., 2023), but are time-consuming. Recent advancements have leveraged feed-forward models with 3D datasets. Some works use diffusion models to generate points (Nichol et al., 2022), neural radiance fields (Wang et al., 2023; Jun & Nichol, 2023; Gupta et al., 2023; Hong et al., 2024), SDF (Cheng et al., 2023; Zhang et al., 2024b), and gaussian splatting (Zhang et al., 2024a). Another line of works uses transformers for auto-regressive generation (Siddiqui et al., 2023; Chen et al., 2024a) or sparse-view reconstruction (Hong et al., 2023; Tang et al., 2024; Zou et al., 2023; Wang et al., 2024a; Xu et al., 2024), which often rely on multi-view diffusion for better performance.

**Multi-View Diffusion Models.** Multi-view models reduce the complexities of 3D synthesis to consistent 2D synthesis. Seminal works (Liu et al., 2023b) have shown novel view synthesis capabilities with pre-trained image diffusion models (Rombach et al., 2022). Later, a plethora of works explored multi-view diffusion models with better consistency (Shi et al., 2023a; Wang & Shi, 2023; Shi et al., 2023b; Long et al., 2023; Liu et al., 2023a) by introducing cross-view communication. More recent works (Voleti et al., 2024; Chen et al., 2024c; You et al., 2024; Han et al., 2024) leverage video priors for multi-view generation by injecting cameras into video diffusion models. However, they still struggle with generalized and controllable generation due to the ill-posed nature of this problem.

**Reference-Augmented Generation.** Retrieval-augmented generation (RAG) emerges to enhance the generation of both language (Lewis et al., 2020) and image (Sheynin et al., 2022; Blattmann et al., 2022) by incorporating relevant external information during the generation process. Under the context of 3D generation, the concept of reference-based generation is also widely applied. Some works (Chaudhuri et al., 2011; Kim et al., 2013; Schor et al., 2019) probe into the database for compatible parts and assemble them into 3D shapes. Some works (Xie et al., 2024; Richardson et al., 2023; Yeh et al., 2024; Zeng et al., 2023; 2024) utilize reference texts or images as guidance to generate textures for existing 3D meshes. Some works refer to a 3D exemplar model (Wu & Zheng, 2022; Wang et al., 2024b) to produce customized 3D assets. Despite success in specific contexts, they are time-consuming with per-case optimization. In contrast, our method focuses on learning a generalized feed-forward model that applies to reference-augmented 3D generation.

## 3 APPROACH

Given one *concept image*, we aim at leveraging an additional *3D reference model* to alleviate 3D inconsistency issues and geometric ambiguity that exist in 3D generation. The 3D reference model can be either provided by the user or retrieved from a large 3D database for different applications.

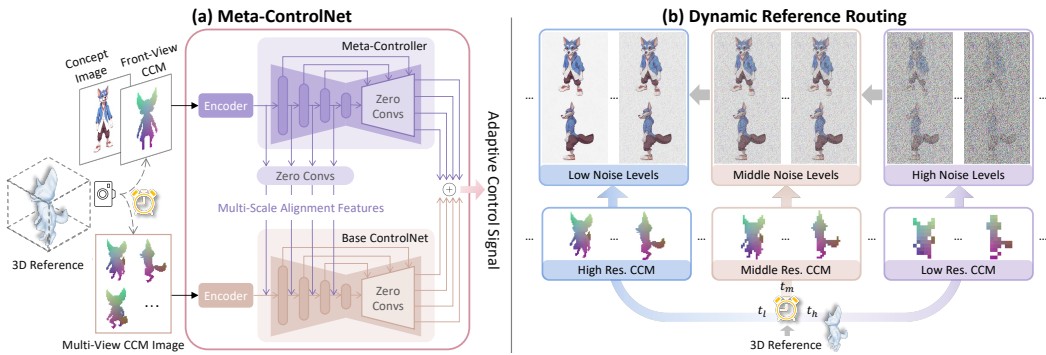

Figure 3: Architectural designs for (a) meta-ControlNet and (b) dynamic reference routing.

The overall pipeline of *Phidias* is shown in Fig. 2, which involves two stages: reference-augmented multi-view generation and sparse-view 3D reconstruction.

## 3.1 REFERENCE-AUGMENTED MULTI-VIEW DIFFUSION

Multi-view diffusion models incorporate camera conditions into well-trained image diffusion models for novel-view synthesis with supervised fine-tuning. We aim to weave additional 3D references into these multi-view models for better generation quality, generalization ability, and controllability. Our approach can be built on arbitrary multi-view diffusion models, enabling reference-augmented 3D content creation from text, image, and 3D conditions. Specifically, we initialize our model with Zero123++ (Shi et al., 2023a), which simply tiles multi-view images for efficient generation conditioned on one input image $c_{\text{image}}$.

To integrate 3D reference models $c_{\text{ref}}$ into the diffusion process, we transform them into multi-view canonical coordinate maps (CCM) (Wang et al., 2024a), also known as NOCS maps (Wang et al., 2019), to condition the diffusion model. The choice of CCMs as the 3D representation is based on two reasons: 1) Multi-view images serve as more efficient and compatible inputs for diffusion models than meshes or voxels, as they have embedded camera viewing angles that correspond with the output images. 2) Reference models often share similar shapes with the concept image but vary significantly in texture details. By focusing on the geometry while omitting the texture, CCMs conditions can reduce generation conflicts arising from texture discrepancies. We add a conditioner branch to incorporate reference CCMs into the base multi-view diffusion model. The objective for training our diffusion model $\epsilon_\theta$ can be then formulated as:

$$\mathcal{L} = \mathbb{E}_{t,\epsilon \sim \mathcal{N}(0,1)} \left[ \| \epsilon - \epsilon_\theta \left( \boldsymbol{x}_t, t, \boldsymbol{c}_{\text{image}}, \boldsymbol{c}_{\text{ref}} \right) \|^2 \right]. \tag{1}$$

To leverage the powerful pertaining capability, only the additional conditioner for reference CCMs is trainable while the base multi-view diffusion is frozen. However, a challenge in our task is that the 3D reference may not strictly align with the concept image or, more commonly, vary in most local parts. We found naive conditioner designs such as ControlNet (Zhang et al., 2023) tend to produce undesirable artifacts, as they were originally designed for image-to-image translation where the generated images strictly align with the condition images. To mitigate this problem, we introduce three key designs for our reference-augmented diffusion model: (1) *Meta-ControlNet* for adaptive control of the conditioning strength (Sec. 3.2); (2) *Dynamic Reference Routing* for dynamic adjustment of the 3D reference (Sec. 3.3); (3) *Self-Reference Augmentation* for self-supervised training (Sec. 3.4).

## 3.2 META-CONTROLNET.

ControlNet is designed to add additional controls to pre-trained diffusion models for image-to-image translation. The conditions are derived from the ground-truth images for self-supervised learning, and thus the generated images are expected to follow the conditions. However, in our settings, the conditions are from the reference model, which often misaligns with the target 3D models we want to generate. The vanilla ControlNet fails to handle such cases. This necessitates further architecture advancement to accordingly adjust conditioning strength when the reference conflicts with the concept image. To this end, we propose meta-ControlNet, as shown in Fig. 3 (a). Meta-ControlNet is comprised of two collaborative subnets, a base ControlNet and an additional meta-controller.

Base ControlNet is comprised of an image encoder, a trainable copy of down-sampling blocks and middle blocks of the base multi-view diffusion, denoted as $\mathcal{F}_\Theta^{\text{base}}(\cdot)$, and a series of $1 \times 1$ zero

convolution layers (Zero Convs) $\mathcal{Z}_\Theta^{\text{base}}(\cdot)$. It takes reference CCM maps $\boldsymbol{c}_{\text{ref}}$ as input to produce the control signal. To deal with misaligned 3D reference, we introduce an additional meta-controller to modulate the conditioning strength according to different similarity levels.

Meta-controller shares a similar architecture but has different parameters $\Theta'$. It works as a knob that dynamically modulates base ControlNet to generate adaptive control signals. Meta-controller takes a pair $\boldsymbol{c}_{\text{pair}}$ of the concept image and the front-view reference CCM as input to produce meta-control signals based on their similarities. The meta-control signals are injected into diffusion models in two ways. On the one hand, meta-controller produces multi-scale alignment features $\boldsymbol{y}_{\text{meta1}} = \mathcal{Z}_{\Theta'}^{\text{meta1}}\left(\mathcal{F}_{\Theta'}^{\text{meta}}\left(\boldsymbol{z}_{\text{pair}}\right)\right)$ to be injected into base ControlNet. These features are applied to the down-sampling blocks of base ControlNet (Eq. 2) at each scale to guide the encoding of reference and help produce base-signals as:

$$\boldsymbol{y}_{\text{base}} = \mathcal{Z}_\Theta^{\text{base}}\left(\mathcal{F}_\Theta^{\text{base}}\left(\boldsymbol{y}_{\text{meta1}}, \boldsymbol{z}_{\text{ref}}\right)\right), \tag{2}$$

where $\boldsymbol{z}_{\text{ref}}$ and $\boldsymbol{z}_{\text{pair}}$ are the feature maps of $\boldsymbol{c}_{\text{ref}}$ and $\boldsymbol{c}_{\text{pair}}$ via the trainable encoders in Fig. 3 (a).

On the other hand, meta-controller produces meta-signals $\boldsymbol{y}_{\text{meta2}} = \mathcal{Z}_{\Theta'}^{\text{meta2}}\left(\mathcal{F}_{\Theta'}^{\text{meta}}\left(\boldsymbol{z}_{\text{pair}}\right)\right)$ to be injected to the pretrained multi-view diffusion models. These features are added up to base-signal $\boldsymbol{y}_{\text{base}}$ to directly apply for the pretrained diffusion models. Totally, the final outputs of meta-ControlNet are adaptive control signals $\boldsymbol{y}_{\text{adaptive}}$ based on the similarity between the concept image and the 3D reference, as:

$$\boldsymbol{y}_{\text{adaptive}} = \lambda(\boldsymbol{y}_{\text{base}} + \boldsymbol{y}_{\text{meta2}}). \tag{3}$$

where $\lambda$ is the coefficient to adjust the strength of control signal for different applications.

### 3.3 Dynamic Reference Routing

Reference models typically align roughly with the concept image in terms of coarse shape, but diverge significantly in local details. This misalignment can cause confusion and conflicts, as the generation process relies on both the image and reference model. To address this issue, we propose a dynamic reference routing strategy that adjusts the reference resolution across denoise timesteps, as shown in Fig. 3 (b). As widely observed during the reverse diffusion process, the coarse structure of a target image is determined in high-noised timesteps and fine details emerge later as the timestep goes on. This motivates us to start with low-resolution reference CCMs at high noise levels $t_h$. By lowering the resolution, reference models provide fewer details but exhibit smaller misalignment with the concept image. This enables reference models to assist in generating the global structure of 3D objects without significant conflicts. We then gradually increase the resolution of reference CCMs as the reverse diffusion process goes into middle noise levels $t_m$ and low noise levels $t_l$ to help refine local structures, *e.g.,* progressively generating a curly tail from a straight one (Fig. 3 (b)). This design choice would ensure effective usage of both concept image and 3D reference during the multi-view image generation process while avoiding degraded generation caused by misalignment.

### 3.4 Self-Reference Augmentation

A good reference model should resemble the target 3D model (with varied details) to provide additional geometric cues, but it is impractical to collect sufficient target-reference pairs for training. An intuitive solution is to retrieve a similar model from a large 3D database as the training reference. However, due to the limited variety in current databases, finding a perfect match is challenging. The retrieved reference can vary greatly in orientation, size and semantics. While this is a common situation in inference scenarios, where a very similar reference is often unavailable, we found training with these challenging pairs fails to effectively use the 3D reference. We conjecture that the learning process struggles due to the significant differences between the reference and target 3D, leading the diffusion model to disregard the references. To avoid the 'idleness' of reference, we developed a self-reference scheme that uses the target model as its own reference by applying various augmentations to mimic misalignment (refer to **Appendix** A.5). This approach ensures that the reference models are somewhat aligned with the target and more compatible, alleviating the learning difficulty. We further design a curriculum training strategy, which begins with minimal augmentations (very similar references) to force the diffusion model to rely on the reference for enhancement. Over time, we gradually increase augmentation strength and incorporate retrieved references, challenging the diffusion model to learn from references that do not closely match the target. Once trained, our model performs well with a variety of references, even those retrieved ones that are not very similar.

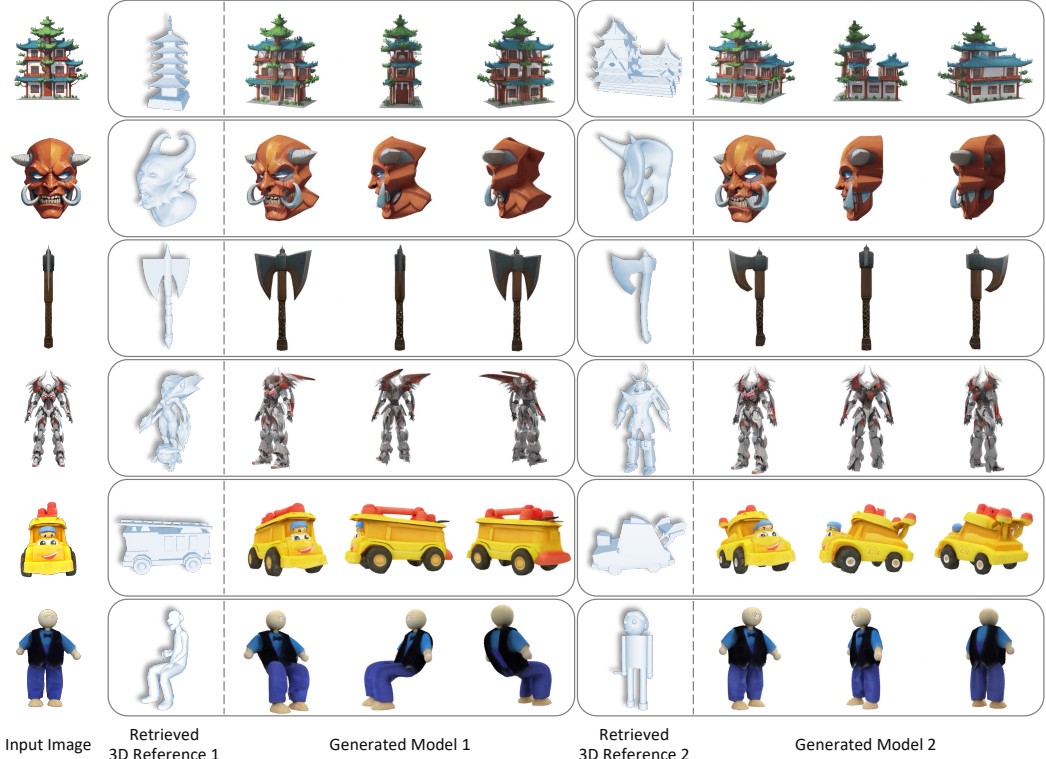

Figure 4: Diverse retrieval-augmented image-to-3D results. With a single input image, *Phidias* can generate diverse 3D models with different 3D references.

## 3.5 SPARSE-VIEW 3D RECONSTRUCTION

With multi-view images generated in the first stage, we can obtain final 3D models via sparse-view 3D reconstruction. This step can be built upon arbitrary sparse-view reconstruction models. Specifically, we finetune LGM (Tang et al., 2024) by expanding the number of input views from 4 to 6 and the resolution of each view from $256 \times 256$ to $320 \times 320$ so that the trained reconstruction model aligns with the multi-view images generated in our first stage.

## 4 EXPERIMENTS

In this section, we evaluate our method on image-to-3D generation, a significant area in 3D generation research. For each image, we retrieve a 3D reference model from a 3D database based on similarity (Zhou et al., 2024). The database used is a subset of Objaverse, containing 40K models. We anticipate that performance could be further enhanced with a larger database in the future. For the rest of this section, we compare *Phidias* with state-of-the-art methods and conduct ablation analysis. More results and implementation details can be found in **Appendix**. Results on text-to-3D and 3D-to-3D generation can be found in Sec. 5.

### 4.1 COMPARISONS WITH STATE-OF-THE-ART METHODS

We compare *Phidias* with five image-to-3D baselines: CRM (Wang et al., 2024a), LGM (Tang et al., 2024), InstantMesh (Xu et al., 2024), SV3D (Voleti et al., 2024), and OpenLRM (He & Wang, 2023).

**Qualitative Results.** *For visual diversity* (Fig. 4), given the same concept image, *Phidias* can generate diverse 3D assets that are both faithful to the concept image and conforming to a specific retrieved 3D reference in geometry. *For visual comparisons* (Fig. 5), while the baseline methods can generate plausible results, they suffer from geometry distortion (*e.g.,* horse legs). Besides, none of the existing methods can benefit from the 3D reference for improved generalization ability (*e.g.,* excavator's dipper) and controllability (*e.g.,* cat's tail) as ours.

**Quantitative Results.** Following previous works, we conduct quantitative evaluation on google scanned objects (GSO) (Downs et al., 2022). We remove duplicated objects with the same shape

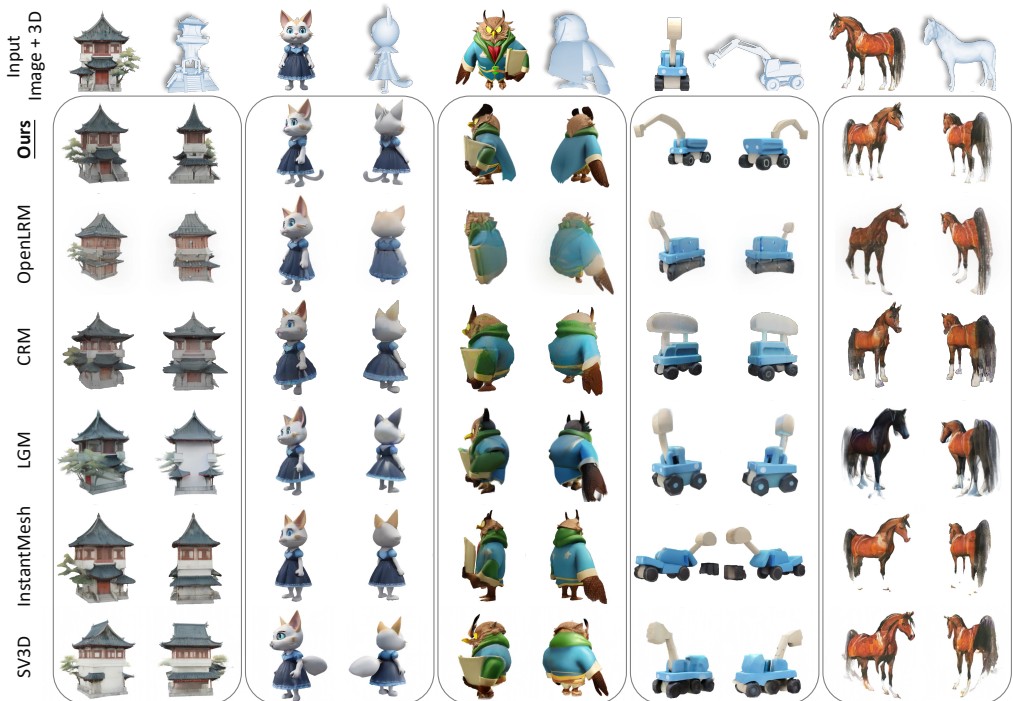

Figure 5: Qualitative comparison on image-to-3D generation.

Table 1: Quantitative comparison with baselines on image-to-3D synthesis.

| Method | PSNR ↑ | SSIM ↑ | LPIPS ↓ | CLIP-P ↑ | CLIP-I ↑ | CD ↓ | F-Score ↑ |
|---|---|---|---|---|---|---|---|
| OpenLRM | 16.15 | 0.843 | 0.194 | 0.866 | 0.847 | 0.0446 | 0.805 |
| LGM | 14.80 | 0.807 | 0.219 | 0.869 | 0.871 | 0.0398 | 0.831 |
| CRM | 16.35 | 0.841 | 0.182 | 0.855 | 0.843 | 0.0443 | 0.796 |
| SV3D | 16.24 | 0.838 | 0.203 | 0.879 | 0.866 | - | - |
| InstantMesh | 14.63 | 0.796 | 0.235 | 0.882 | 0.880 | 0.0450 | 0.788 |
| **Ours (GT Ref.)** | **20.37** | **0.870** | **0.117** | **0.911** | **0.885** | **0.0391** | **0.840** |
| **Ours (Retrieved Ref.)** | 17.02 | 0.845 | 0.174 | 0.887 | 0.885 | 0.0402 | 0.833 |

and randomly select 200 objects for evaluation. *For visual quality*, we report reconstruction metrics (PSNR, SSIM and LPIPS) on 20 novel views. We also report novel views' CLIP similarity with paired GT (CLIP-P) and input image (CLIP-I). *For geometry quality*, we sample 50K points from mesh surface and compute Chamfer Distance (CD) and F-Score (with a threshold of 0.05). To align the generated mesh and GT, we unify their coordinate systems and re-scale them into a unit box. We report our results with the retrieved reference, *i.e., Ours (Retrieved Ref.)*, and GT mesh as reference, *i.e., Ours (GT Ref.)*, respectively. As shown in Tab. 1, ours, with either retrieved or GT reference, outperforms all baselines, benefiting from the proposed retrieval-augmented method. The results of *Ours (Retrieved Ref.)* seems marginal although *Phidias* produces plausible 3D models given various references that differ from GT (Fig. 7 (a)). We argue that this is caused by the differences between the retrieved references and GT when computing the reconstruction metrics. The actual performance of *Phidias* should be between *Ours (GT Ref.) and Ours (Retrieved Ref.)*.

**User Study.** We further conduct a user study to evaluate human preferences among different methods. We publicly invite 30 users to complete a questionnaire for pairwise comparisons. We show the preference rate (*i.e.,* the percentage of users prefer ours compared to a baseline method) in Tab. 2, which suggests that our approach significantly outperforms existing methods in the image-to-3D task based on human preferences.

## 4.2 ABLATION STUDY AND ANALYSIS

**Ablation Studies.** We conduct ablation studies across four settings: a base model employing a standard ControlNet trained with self-reference, and three variants (each integrating one proposed component into the base model). The quantitative results in Tab. 3 demonstrate clear improvements in both visual and geometric metrics with our proposed components.

Table 2: User study.

| Baseline | Pref. Rate |
|----------|------------|
| OpenLRM | 94.7% |
| LGM | 95.8% |
| CRM | 93.7% |
| SV3D | 88.4% |
| InstantMesh | 91.6% |

Table 3: Quantitative ablation study of the proposed components.

| Method | PSNR ↑ | SSIM ↑ | LPIPS ↓ | CLIP-P ↑ | CLIP-I ↑ | CD ↓ | F-Score ↑ |
|--------|--------|--------|---------|----------|----------|------|-----------|
| Base Model | 14.70 | 0.804 | 0.227 | 0.855 | 0.859 | 0.0424 | 0.826 |
| + Meta-ControlNet | 16.35 | 0.833 | 0.190 | 0.881 | 0.878 | 0.0407 | 0.829 |
| + Dynamic Ref. Routing | 14.76 | 0.816 | 0.221 | 0.868 | 0.861 | 0.0420 | 0.826 |
| + Self-Ref. Augmentation | 16.57 | 0.840 | 0.182 | 0.880 | 0.883 | 0.0414 | 0.830 |
| Full Model | **17.02** | **0.845** | **0.174** | **0.887** | **0.885** | **0.0402** | **0.833** |

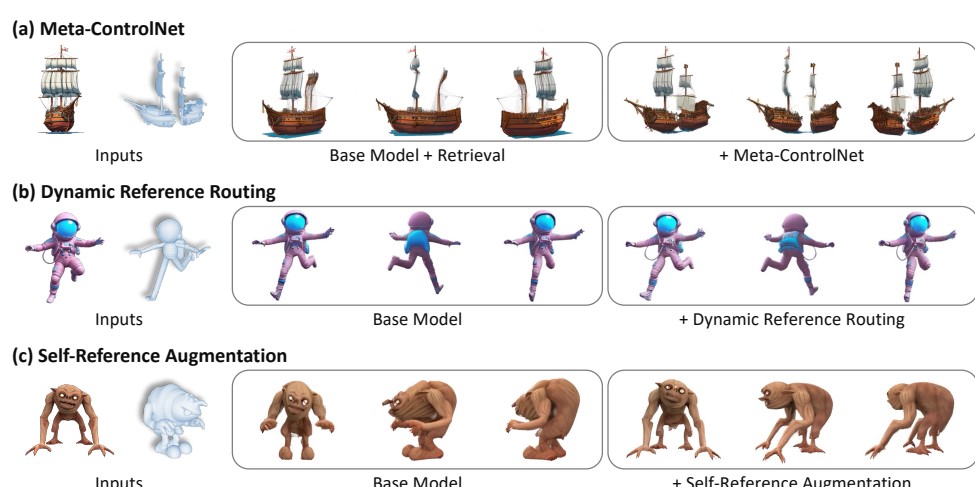

**(a) Meta-ControlNet**

Inputs      Base Model + Retrieval      + Meta-ControlNet

**(b) Dynamic Reference Routing**

Inputs      Base Model      + Dynamic Reference Routing

**(c) Self-Reference Augmentation**

Inputs      Base Model      + Self-Reference Augmentation

Figure 6: Qualitative ablation study of the proposed components.

*Effectiveness of Meta-ControlNet.* To evaluate meta-ControlNet, we use both self-reference and retrieved reference for training, as the learning of Meta-Controller (Fig. 3 (a) top) requires reference models with varying levels of similarity. As shown in Fig. 6 (a), the base model trained with retrieved reference often ignores the reference, failing to follow the shape pattern (disconnected boat). This phenomenon stems from the considerable similarity variation among retrieved references, which confuses the diffusion model. The base model thereby struggles to determine when and how to use the reference as it lacks the ability to adjust to different levels of similarity. Consequently, they often end up with ignoring the reference models entirely. In contrast, meta-ControlNet equips the model with the capability to dynamically modulate the conditioning strength of the reference model, thereby effectively utilizing available references for improving or controlling the generation process.

*Effectiveness of Dynamic Reference Routing.* Dynamic reference routing aims to alleviate local conflicts between the reference and concept images. As illustrated in Fig. 6 (b), when given a highly similar reference, the base model tends to rely heavily on it, leading to missing specific local details within the concept image, *e.g.,* the rope on the left. By addressing these conflicts with dynamic routing, the model maintains the essential details of the concept image, while still benefiting from the guidance of the 3D reference.

*Effectiveness of Self-Reference Augmentation.* As shown in Fig. 6 (c), without self-reference augmentation, the base model predominantly depends on the provided reference for generation. When given a significantly misaligned reference, the model tends to follow the reference's structure, resulting in an undesired outcome. Conversely, self-reference augmentation ensures that the generated models remain faithful to the concept image, while using the reference as geometry guidance.

**Analysis on Similarity Levels of the 3D Reference.** We analyze how similarity levels of 3D references would affect the performance. For each input, we retrieve three models ranked first (top-1), third (top-3), and fifth (top-5) in similarity scores, and randomly choose one model, to serve as 3D references. Quantitative results in Tab. 4 indicate that *Phidias* performs better with more similar references. Fig. 7 (a) shows *Phidias* generates diverse plausible results with different references. All results remain faithful to the input image in the front view, but show variations in shapes influenced by the specific reference used. Also, we found *Phidias* can still generate plausible results even with a random 3D reference (Fig. 7 (a) middle), indicating robustness to reference with different similarity levels. Moreover, as shown in Fig. 7 (b), *Phidias* will ignore inappropriate 3D reference with explicit conflicts and generate satisfying results following the input image, demonstrating the robustness to some extreme cases when the retrieved 3D reference significantly differs from the image.

Table 4: Quantitative analysis on similarity levels of the 3D reference.

| Reference | PSNR ↑ | SSIM ↑ | LPIPS ↓ | CLIP-P ↑ | CLIP-I ↑ | CD ↓ | F-Score ↑ |
|---|---|---|---|---|---|---|---|
| Top-1 Retrieval | **17.02** | **0.845** | 0.174 | **0.887** | 0.885 | 0.0402 | **0.833** |
| Top-3 Retrieval | 16.75 | 0.841 | **0.172** | **0.887** | **0.886** | **0.0395** | 0.830 |
| Top-5 Retrieval | 15.96 | 0.835 | 0.185 | 0.886 | 0.884 | 0.0408 | 0.819 |
| Random Reference | 14.74 | 0.820 | 0.226 | 0.884 | 0.882 | 0.0424 | 0.810 |
| Without Reference | 15.90 | 0.836 | 0.188 | 0.886 | 0.880 | 0.0416 | 0.814 |

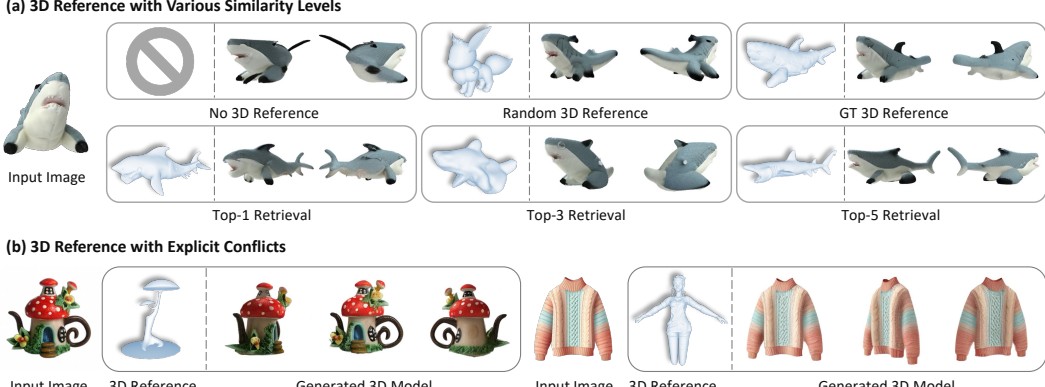

Figure 7: Qualitative analysis on similarity levels of the 3D Reference.

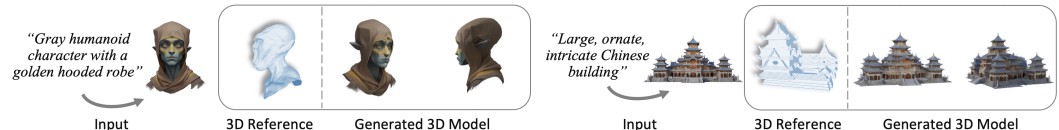

Figure 8: *Phidias* enables retrieval-augmented text-to-3D generation by first converting the input text into a concept image, and then retrieving a 3D reference based on both the text and image.

# 5 APPLICATIONS

*Phidias* supports versatile applications beyond image-to-3D, such as text-to-3D, theme-aware 3D-to-3D, interactive 3D generation with coarse guidance, and high-fidelity 3D completion.

**Text to 3D.** Text-to-3D generation can be converted to image-conditioned generation by transforming a text prompt into a concept image. However, the generated concept image can sometimes be atypical and may lose some information compared with original text input. To enhance generative quality, *Phidias* employs retrieval-augmented text-to-3D generation, as illustrated in Fig. 8. This involves first retrieving a set of 3D references based on the concept image, and then selecting the one that most closely matches the text description as the final reference.

**Theme-Aware 3D-to-3D Generation.** This task aims to create a gallery of theme-consistent 3D variations from existing 3D models. Previous work (Wang et al., 2024b) proposed an optimization-based approach, which is time-consuming. *Phidias* supports fast generation by first generating image variations based on the input 3D model, and then transforming these variant images into 3D variations with the original 3D model itself as reference. The results are shown in Fig. 9, using 3D models from Sketchfab and previous works as inputs.

**Interactive 3D Generation with Coarse Guidance.** Interactive generation gives users more control over the outputs, empowering them to make quick edits and receive rapid feedback. *Phidias* also provides this functionality, allowing users to continually adjust the geometry of generated 3D models using manually created coarse 3D shapes as reference models, as shown in Fig. 10.

**High-Fidelity 3D Completion.** Given incomplete 3D models, as shown in Fig. 11, *Phidias* can be used to restore the missing components. Specially, by generating a complete front view through image inpainting and referencing to the original 3D model, *Phidias* can precisely predict and fill in the missing parts in novel views while maintaining the integrity and details of the origin, resulting in a seamlessly and coherently structured 3D model.

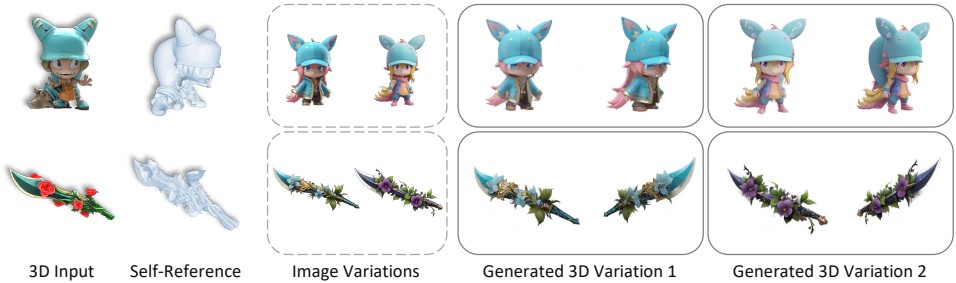

Figure 9: *Phidias* facilitates rapid, theme-aware 3D-to-3D generation by using an existing 3D model as a reference to transform its image variations into corresponding 3D variations.

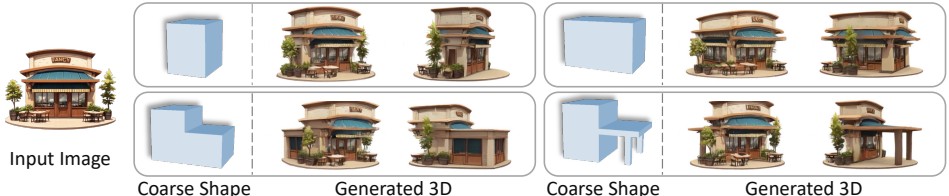

Figure 10: *Phidias* enables interactive 3D generation with coarse 3D shapes as guidance.

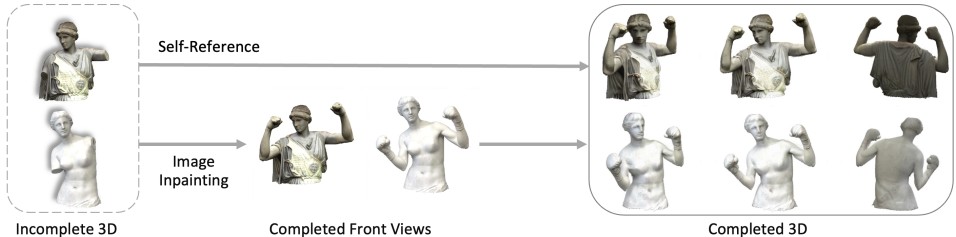

Figure 11: *Phidias* supports high-fidelity 3D completion by using the completed front views to guide the restoration of the missing parts and the original 3D model to help preserve the original details.

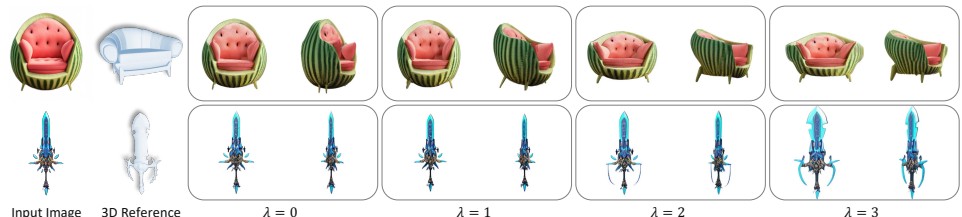

Figure 12: *Phidias* allows users to explicitly decide the reference strength during inference.

**3D RAG with Adjustable Reference Strength.** *Phidias* offers users enhanced control over the reference strength by adjusting $\lambda$ in Eq. 3. This feature works as a "control knob" to vary the emphasis on the reference, which is particularly useful for scenarios like image-based 3D re-texturing and interpolation between the concept and reference, as illustrated in Fig. 12.

## 6 CONCLUSION

In this work, we introduce *Phidias*, a 3D-aware diffusion model enhanced by 3D references. By incorporating meta-ControlNet, dynamic reference routing, and self-reference augmentations, *Phidias* effectively leverages reference models with varying degrees of similarity for 3D generation. The proposed approach boosts the quality of 3D generation, expands its generalization capabilities, and improves user control. *Phidias* offers a unified framework for creating high-quality 3D content from diverse modalities, which enables versatile applications beyond image-to-3D, such as, but not limited to, text to 3D, theme-aware 3D-to-3D generation, interactive 3D generation, and high-fidelity 3D completion. More discussions of limitations and failure cases can be found in **Appendix B**.

ACKNOWLEDGMENTS

This work is partially supported by the National Key R&D Program of China (2022ZD0160201) and Shanghai Artificial Intelligence Laboratory. This work is also in part supported by a GRF grant from the Research Grants Council of Hong Kong (Ref. No.: 11205620).

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

APPENDIX

# A  IMPLEMENTATION DETAILS

## A.1  DATASET

**Training set.** To train our *reference-augmented multi-view diffusion model*, we use a filtered subset of the Objaverse (Deitke et al., 2023) dataset, excluding low-quality 3D models as described in (Tang et al., 2024). Additionally, we apply further filtering to remove objects that are too thin and eliminate data originating from scans, both of which are intended to ensure the quality of subsequent retrieval. We also exclude objects with an excessively high number of vertices or faces to optimize the costly point cloud extraction process and reduce computational time. These refinements result in a final training set comprising approximately 64K 3D objects. For each object, we normalize it within a unit sphere, and render 1 concept image, 6 canonical coordinate maps (CCMs), and 6 target RGBA images, following the camera distribution protocol of Zero123++ (Shi et al., 2023a). In particular, the concept image is rendered using randomly sampled azimuth and elevation angles from a predefined range. The poses of the six corresponding CCMs and target images consist of interleaving absolute elevations of $\{20°, -10°, 20°, -10°, 20°, -10°\}$, and relative azimuths of $\{\phi + 30°, \phi + 90°, \phi + 150°, \phi + 210°, \phi + 270°, \phi + 330°\}$, where $\phi$ represents the azimuth of the concept image. To train our *sparse-view 3D reconstruction model*, we adopt the same training set and render images from 32 randomly sampled camera views. All images are rendered at a resolution of $512 \times 512$, a fixed absolute field of view (FOV) of $30°$, and a fixed camera distance of $1.866$.

**Retrieval data and method.** We leverage Uni3D (Zhou et al., 2024) to retrieve a 3D reference from an input image. In Uni3D, the latent space of the point cloud encoder is aligned to the OpenCLIP (Ilharco et al., 2021) image embedding space, facilitating seamless image-to-PointCloud retrieval. Before retrieval, point clouds are sampled from meshes according to the probability distribution of face areas, ensuring denser sampling in regions with larger surface areas. Each point cloud contains 10K points. As point cloud preprocessing is time-consuming, we limit our retrieval to a subset of 40K objects from Objaverse. Our retrieval database contains precomputed embeddings generated by the Uni3D point cloud encoder, which are compared with the query vector of an input image using cosine similarity. To obtain the query vector, we first apply normalization transforms to align the input image with the pre-trained EVA02-E-14-plus model from OpenCLIP, which acts as the query encoder. The normalized image is then encoded into a feature vector. The top candidates are selected based on the highest similarity scores, and a softmax function is applied to the top-k scores to enable probabilistic sampling, ensuring efficient and accurate matching between the input image and the corresponding point clouds.

## A.2  TRAINING

**Reference-augmented multi-view diffusion model.** *White-Background Zero123++.* As discussed in Sec. 3.1, we select Zero123++ as our initial multi-view diffusion model. Upon receiving an input image, Zero123++ generates a tailored multi-view image at a resolution of $960 \times 640$, comprising six $320 \times 320$ views arranged in a $3 \times 2$ grid. The original Zero123++ produces images with a gray background, which can result in floaters and cloud-like artifacts during the subsequent sparse-view 3D reconstruction phase. To mitigate this issue, we initialize our model with a variant of Zero123++ (Xu et al., 2024), which is finetuned to generate multi-view images with a white background.

*Training Details.* During the training of our reference-augmented multi-view diffusion model, we use the rendered concept image and six CCMs of a 3D object as conditions, and six corresponding target images tailored to a $960 \times 640$ image as ground truth image for denoising. All images and CCMs have a white background. We concatenate the concept image and the front-view CCM along the RGB channel as the input for meta-ControlNet. For the proposed dynamic reference routing, we dynamically downsample the original CCMs to lower resolutions and then upsample them to $320 \times 320$, using the nearest neighbor. Specifically, we start with a resolution of 16 at noise levels of $[0, 0.05)$ and gradually increase the resolution to 32 and 64 at noise levels of $[0.05, 0.4)$ and $[0.4, 1.0]$, respectively. For self-reference augmentations (Sec. A.5), the probabilities of applying random resize, flip horizontal, grid distortion, shift, and retrieved reference are set to 0.4, 0.5, 0.1, 0.5, and 0.2, respectively. We train the model for 10,000 steps, beginning with 1000 warm-up steps

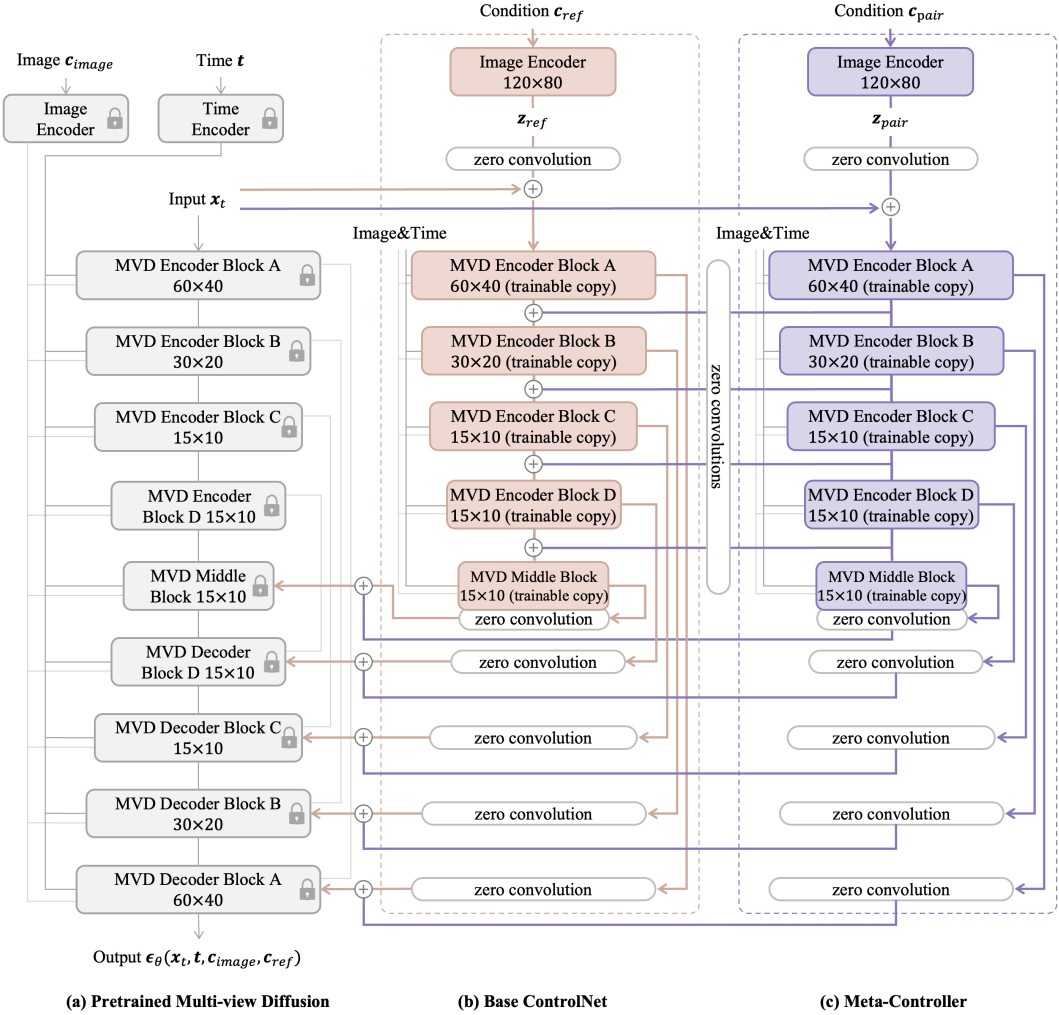

Figure 13: Detailed architecture design of meta-ControlNet.

with minimal augmentations. We use the AdamW optimizer with a learning rate of $1.0 \times 10^{-5}$ and a total batch size of 48. The whole training process takes around 10 hours on 8 NVIDIA A100 (80G) GPUs.

**Sparse-view 3D reconstruction model.** As discussed in Sec. 3.5, we employ LGM to convert the synthesized multi-view images into a 3D model. The original LGM is designed to reconstruct a 3D model from four input views at a resolution of $256 \times 256$. However, this does not align with the multi-view images generated in our first stage, which consist of six views at a resolution of $320 \times 320$. To adapt LGM to our specific inputs, we take its pretrained weights as initialization and finetune it to support six input images at $320 \times 320$. Simultaneously changing the number of input views and image resolutions can destabilize the training process. We therefore separate the finetuning of number of input views and input resolution. Specifically, we first finetune the model with six input views at the original resolution for 60 epochs and then further finetune the model at a higher resolution of $320 \times 320$ for another 60 epochs. The finetuning process is conducted on 32 NVIDIA A100 (80G) GPUs using the AdamW optimizer with a learning rate of $2.0 \times 10^{-4}$ and a total batch size of 192. The whole finetuning process takes around four days.

## A.3 INFERENCE SPEED

Similar to existing image-to-3D generation methods Wang et al. (2024a); Xu et al. (2024); Tang et al. (2024) that combine multi-view generation and sparse-view 3D reconstruction, *Phidias* can generate a final 3D model from the input image in less than 10 seconds. The major latency arises from the diffusion-based multi-view generation process, which takes 9 seconds to generate an image with 6 views in a $3 \times 2$ grid by 75 denoising steps. Other inference-related processes are nearly real-time,

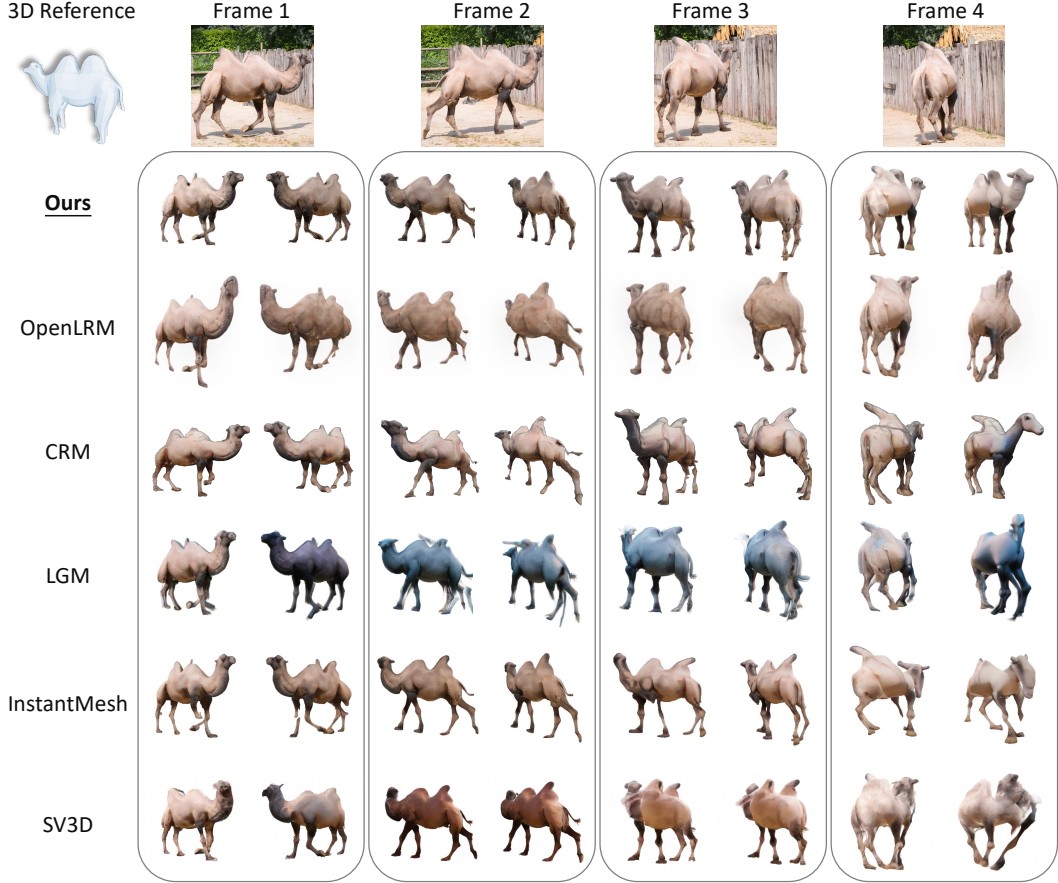

Figure 14: Analysis on different input viewpoints. We compare the performance of *Phidias* with five baseline methods by reconstructing 3D objects from video frames with various viewpoints. For each case, we show two rendered images at novel views.

including 0.03 seconds for image-based 3D retrieval, 0.4 seconds for rendering the CCM maps, and 0.02 seconds for sparse-view 3D reconstruction. The inference time is tested using a single NVIDIA A100 GPU.

### A.4 META-CONTROLNET

A detailed figure of the proposed meta-ControlNet in the style of vanilla ControlNet is shown in Fig. 13, where $c_{pair}$ is a pair of the concept image and the front-view reference CCM.

### A.5 AUGMENTATION DETAILS

We implement a series of augmentations to facilitate the training of our diffusion model in a self-reference manner, where the ground truth 3D model serves as its own reference. These augmentations are designed to simulate the misalignment between the 3D reference and the concept image.

*Resize and horizontal flip.* Due to the self-reference strategy, reference CCMs are always pixel-wise aligned with the concept image. However, during inference, references often differ in scale or exhibit mirror symmetry. For example, a reference 3D character might hold a weapon in the opposite hand compared to the concept image. To address this, we apply random resizing and horizontal flipping to the reference model, simulating scale variations and mirror-symmetric structures.

*Grid distortion and shift.* During inference, the reference may exhibit asymmetric similarity with the target 3D model across different views. For instance, a reference building might closely resemble

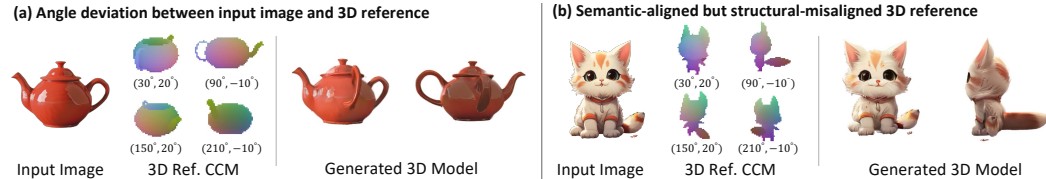

Figure 15: Failure cases. There are two typical failure cases due to bad retrieval: (a) misaligned pose and (b) misaligned structure.

the concept image from the front but differ significantly from the side. To address this, we apply multi-view jitter through grid distortion and shifting. Specifically, we independently distort and shift each view of the reference CCMs using a random grid and a random shift offset during training, simulating such asymmetric similarity across views.

*Retrieved Reference.* Although the retrieved 3D reference alone is insufficient for model training, as discussed in Sec. 3.4, it can still serve as a strong augmentation to simulate significant misalignment. Therefore, we assign a small probability of using the retrieved model as the reference during training.

## B  LIMITATION AND FAILURE CASES

Despite promising results, *Phidias* still has several limitations for further improvement. As a retrieval-augmented generation model, the performance can be affected by the retrieval method and the scale and quality of 3D reference database. Currently, the 3D database we used for retrieval only consists of 40K objects, making it difficult to find a very similar match. Also, mainstream 3D retrieval methods (Zhou et al., 2024; Xue et al., 2023) rely on semantic similarity, which may not always yield the best match. For example, retrieved reference models with misaligned poses or structures can lead to undesired outcomes, as shown in Fig. 15. Future works that improve the retrieval accuracy and expand the 3D reference database could mitigate these issues. Additionally, the limited resolution of the backbone multi-view diffusion model ($320 \times 320$) restricts the handling of high-resolution images. Enhancing the resolution of the diffusion model could further improve the quality of the generated 3D models.

## C  ADDITIONAL RESULTS

### C.1  ADDITIONAL ANALYSIS ON ENHANCED GENERALIZATION ABILITY

*Phidias* takes an additional 3D reference as input to improve generative quality (Fig. 5) and provide greater controllability (Fig. 4) for 3D generation. We argue that *Phidias* can also enhance generalization ability when given input images from atypical viewpoints. When reconstructing 3D objects from video frames with varying views (Fig. 14), we observe that the baseline methods perform well with typical view angles (*i.e.,* frame 1) but struggle with atypical input view angles (*e.g.,* frame 3 and 4). Conversely, *Phidias* produces plausible results given all four input views, demonstrating robust generalization ability across both typical and atypical viewpoints.

### C.2  MORE RESULTS

More results on theme-aware 3D-to-3D generation are shown in Fig. 16. More results on text-to-3D and image-to-3D generation are shown in Fig. 17 and Fig. 18.

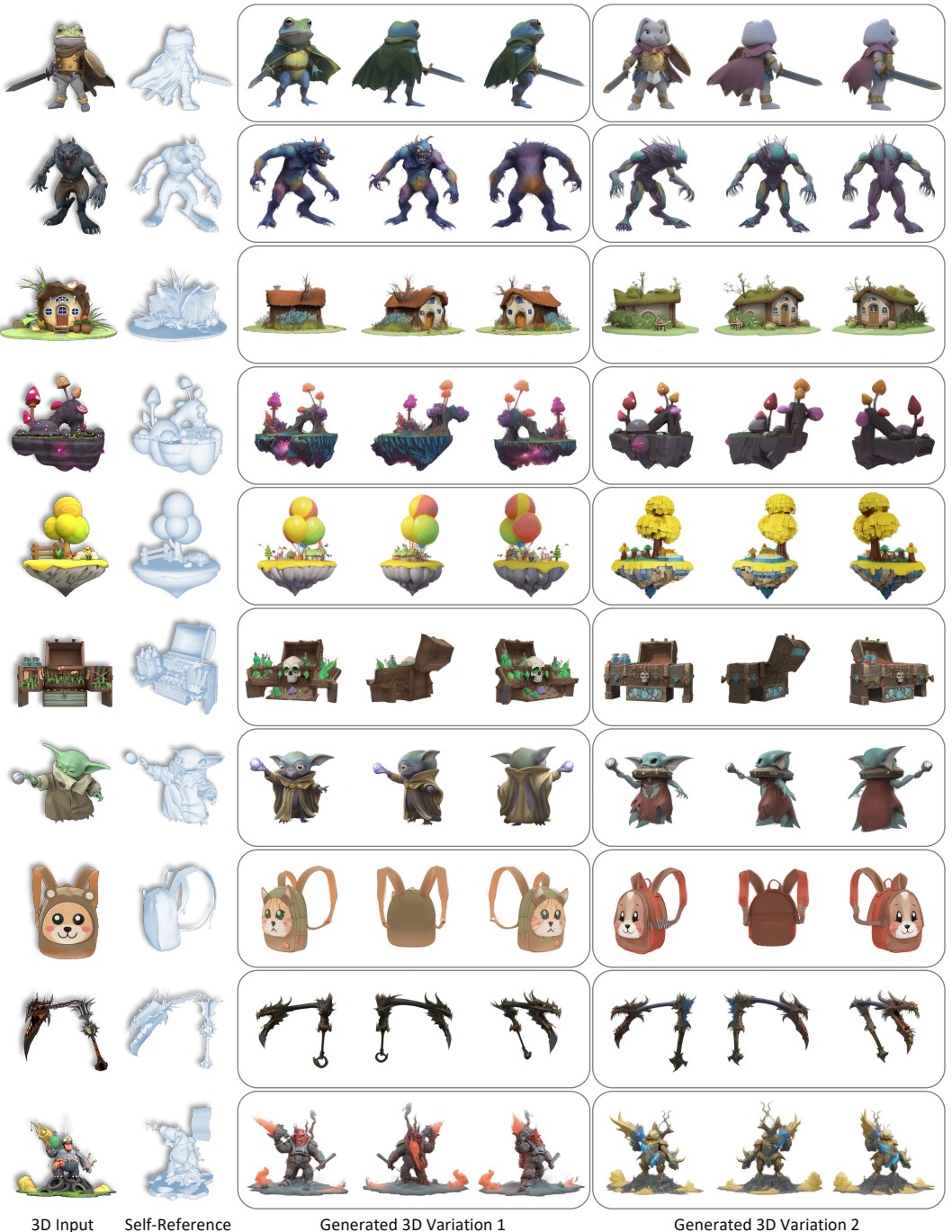

Figure 16: Additional results on theme-aware 3D-to-3D generation.

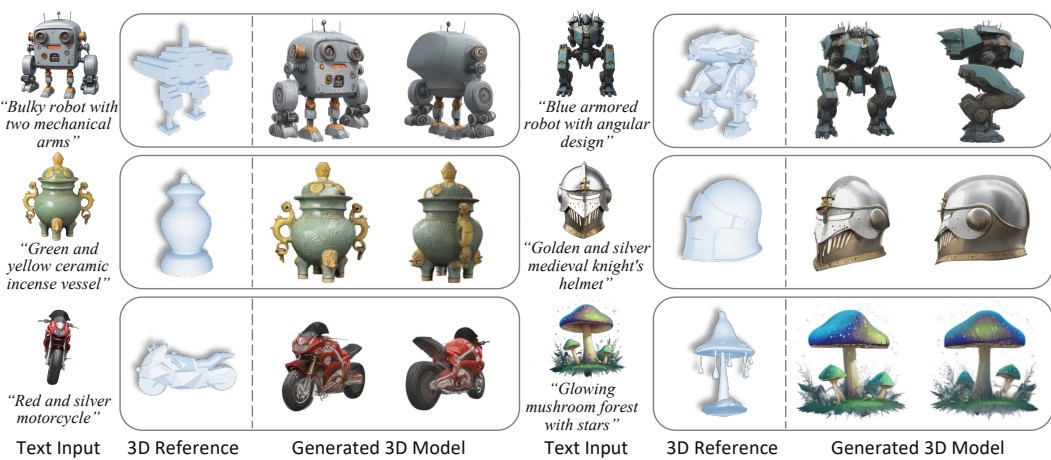

Figure 17: Additional results on retrieval-augmented text-to-3D generation.

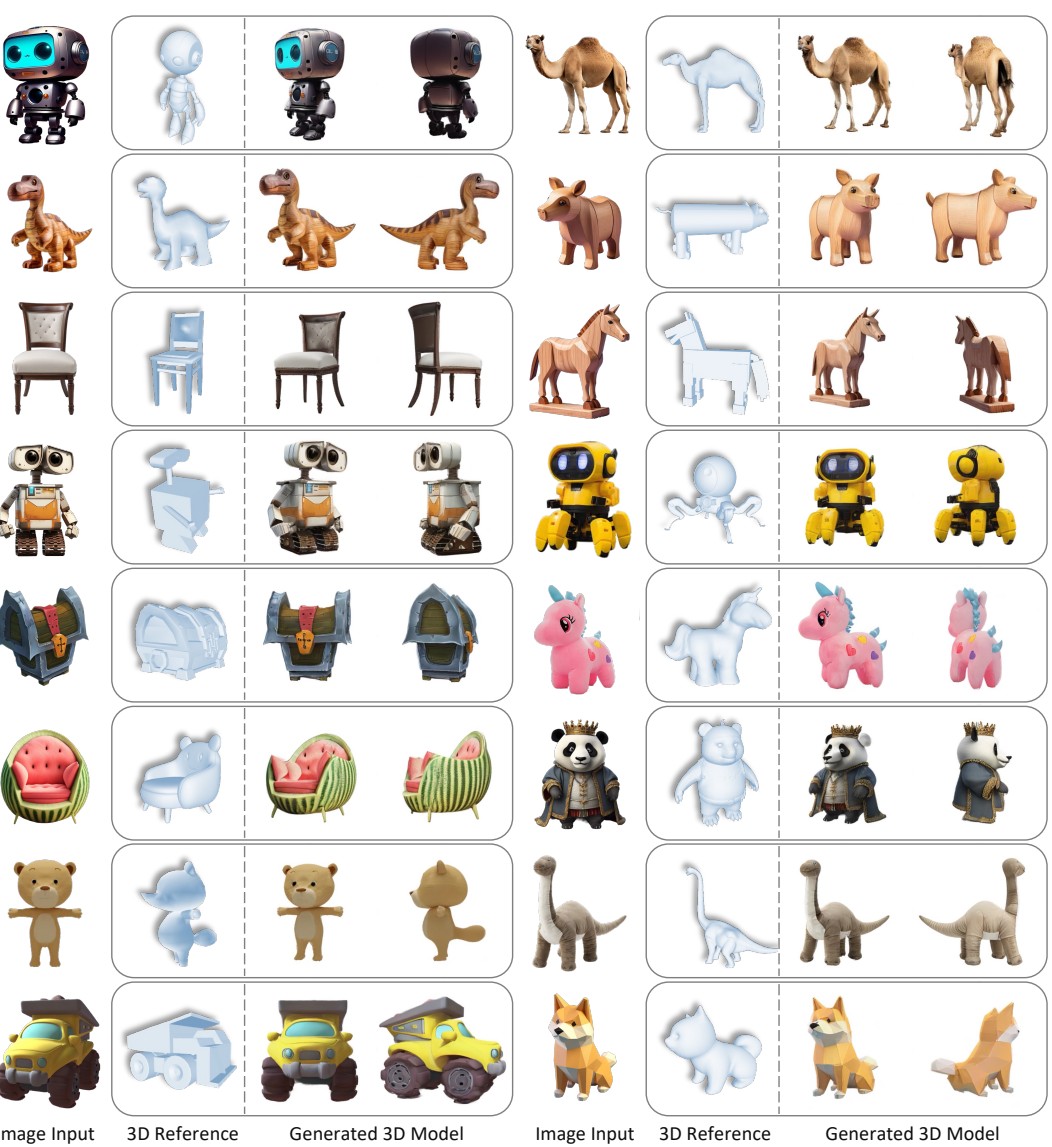

Figure 18: Additional results on retrieval-augmented image-to-3D generation.

