# OpenReview forum: "Phidias: A Generative Model for Creating 3D  Content from Text, Image, and 3D Conditions with Reference-Augmented  Diffusion"
_ICLR.cc/2025/Conference — ICLR 2025 Poster_

### Official Review · Reviewer_Ef4X · 2024-10-16

**Soundness:** 3
**Presentation:** 3
**Contribution:** 3
**Rating:** 8
**Confidence:** 4

**Summary:**

This paper proposes a new 3D generation method that introduces a meta-ControlNet and dynamic reference routing, leveraging user-provided 3D reference models and images to produce high-quality, controllable 3D object synthesis.

**Strengths:**

- The introduction of meta-ControlNet, combined with 3D reference models, provides strong guidance for the generation process, resulting in high-quality 3D outputs with excellent geometric consistency.

- As a reviewer familiar with this field, I believe that the combination of 3D reference models and images for generating 3D results has significant industrial value and could greatly inform future work in 4D generation as well.

- The method is simple, easy to understand, and the paper is clearly written.

**Weaknesses:**

- Obtaining a reference 3D model can be quite challenging. Although this is valuable in certain application scenarios, it can be difficult to find appropriate reference 3D models in extreme cases. I recommend that the authors add this limitation to the paper.

- While the strategy of combining ControlNet for generation appears straightforward, this could be criticized for lacking novelty. However, in my opinion, simplicity often leads to elegant and effective solutions, and I don’t consider this a reason for rejection.

**Questions:**

- I strongly support the task of combining reference 3D models with images for 3D generation as proposed in the paper, especially considering its relevance to the gaming industry and asset creation. Some related 3D generation methods discuss a texture editing task using complex images to guide coarse 3D object generation, which is highly relevant to this work. Could the authors provide some discussion or analysis on how to improve the method when there are significant differences between the reference image and the reference 3D model? This could be addressed in future work.

- I am inclined to accept this paper, as I also work on related tasks. Currently, I rate it as 6 but would likely increase the rating to 8 during the rebuttal phase.

---

> ### Author Response · Authors · 2024-11-20
> **Response to R-Ef4X**
>
> Dear Reviewer Ef4X,
>
> Thanks for your careful review and constructive comments. We appreciate you recognizing our paper as an impactful work with significant industrial value. We note that you still have several concerns about the performance of Phidias in extreme cases and the novelty of the proposed solution. We would like to provide more details regarding your questions and hope your concerns can be addressed.
>
> **W1: Limitation of inappropriate 3D reference in extreme cases.**
>
> We have already discussed this limitation in Appendix (B). Note that when the 3D reference significantly conflicts with the concept image, our performance still equals the existing SOTA image-to-3D generation method, as shown in Fig. 7 (b). Please refer to the 2nd point of our Common Response above for more details.
>
> **W2: Combining ControlNet could be criticized for lacking novelty.**
>
> As you said, "Simplicity often leads to elegant and effective solutions". While the combination of ControlNet for reference-augmented multi-view generation seems straightforward, it is not that easy to adapt ControlNet to our task due to the Misalignment Dilemma, where the discrepancy between the concept image and the 3D reference can lead to conflicts in the generation process (L78-80). Our novelty comes from introducing the novel task of RAG-based 3D generation, our specific architecture designs for tackling misalignment dilemmas, and extending our framework for unified 3D generation from multi-modality inputs with various useful applications.
>
> **Q1: How can we improve the method when there are significant differences?**
>
> This depends on the user’s intention.
>
> - Currently, for cases that emphasize image more than the 3D reference (the default setting of Phidias), such as image-to-3D generation, Phidias will follow the concept image and ignore the 3D reference if there are significant differences between them, as shown in Fig. 7 (b).  Please refer to the 2nd point of our Common Response above for more details.
>
> - Currently, for cases where users want to decide which side to follow most, Phidias allows users to adjust the reference strength manually using a “control knob”, as shown in Fig. 12 and L483.
>
> - In the future, if we use the native 3D diffusion model for the RAG-based 3D generation, we may be able to leverage the information of all kinds of 3D references even if they are irrelevant to the image during inference, by utilizing their spatial structure/distribution (sparse or dense) and number of faces as conditions.
>
> **Q2: Increase the rating**
>
> Again, thank you for recognizing the value of our paper. We hope our responses above address your concerns about our paper, and it would be appreciated if there is an increased rating. We would also like to hear more from you if you have further questions or comments.
>
> Bests,
>
> Authors

---

> ### Comment · Reviewer_Ef4X · 2024-11-24
> **Response to authors**
>
> We sincerely thank the authors for their detailed response. For using complex images to edit the texture of 3D objects, the following articles could be helpful:
>
> [1] StyleTex: Style Image-Guided Texture Generation for 3D Models \
> [2] Texture: Text-guided texturing of 3D shapes \
> [3] TextureDreamer: Image-guided Texture Synthesis through Geometry-aware Diffusion \
> [4] Ipdreamer: Appearance-controllable 3D object generation with image prompts \
> [5] Paint3D: Paint Anything 3D with Lighting-Less Texture Diffusion Models
>
> There is no need to include additional experiments. Instead, a simple analysis of the similarities and differences between your paper and these methods would suffice. Additionally, consider citing these articles. A concise response would be appreciated, and I will raise my rating accordingly.

---

> > ### Author Response · Authors · 2024-11-24
> >
> > Hi Reviewer Ef4X,
> >
> > We sincerely thank you for your response and constructive comments/suggestions. As suggested, we have cited these articles as part of our related work in L149-151.
> >
> > The **similarity** is that both these works and ours use an existing 3D mesh and an image as input to generate a textured 3D object as output. However, their task is not aligned with ours. The **difference** is that these works aim to keep the geometry of the input 3D mesh unchanged and only generate texture based on a reference image. In contrast, our task is image-to-3D generation, which aims to create both geometry and texture based on the input image. The input 3D mesh in our task is only an auxiliary guidance to improve quality and provide additional controllability in novel views, where the geometry of the final generated 3D object can be different from the input 3D.
> >
> > We hope our responses above address your concerns about our paper. We would also like to hear more from you if you have further questions or comments.
> >
> > Best regards,
> >
> > Authors

---

> > > ### Comment · Reviewer_Ef4X · 2024-11-24
> > > **Response to Authors**
> > >
> > > We sincerely thank you for your response, good luck for you.

---

### Official Review · Reviewer_eNa9 · 2024-10-22

**Soundness:** 3
**Presentation:** 3
**Contribution:** 3
**Rating:** 6
**Confidence:** 4

**Summary:**

This paper introduces Phidias, a RAG-based framework for high-fidelity 3D content creation from multimodal inputs, including text, images, and 3D references. The framework introduces key innovations include Meta-ControlNet to dynamically modulate reference influence, dynamic reference routing to progressively refine outputs through a coarse-to-fine strategy, and self-reference augmentation to enhance generalization. Phidias enables sparse-view reconstruction, theme-based model variation, and user-interactive generation, demonstrating clear superiority over state-of-the-art approaches in both geometric precision and visual fidelity.

**Strengths:**

1. This paper presents a well-founded solution for multi-view generation, effectively addressing challenges such as the generation of unseen views and other ill-posed problems.
2. This paper is well-organized and clearly written, making it accessible to readers.

**Weaknesses:**

1. A significant weakness is the limited analysis of the retrieval mechanism for 3D references, which plays a critical role in guiding generation. The reliance on existing 3D models introduces a potential bottleneck, as Phidias’s performance may degrade if the retrieved references are irrelevant or of poor quality.
2. The control mechanism introduced via Meta-ControlNet is promising, but the paper does not delve deeply into more challenging settings. For example, scenarios where the input concept conflicts with the reference.

**Questions:**

1. why the point cloud for retrieval only contains 10K points, is it limitation stems from constraints imposed by Uni3D?
2. Why are the performences in Figure 14 so different from the results of the last row of figures in Figure 4? It looks like that some case's pose in Figure 4 doesn't match the input image, either.
3. The time and resource consumption for retrieval during each inference is not clearly stated

---

> ### Author Response · Authors · 2024-11-20
> **Response to R-eNa9**
>
> Dear Reviewer eNa9,
>
> Thanks for your constructive comments.  We hope our responses below address your concerns about our paper. We kindly invite you to watch the attached demo video in the supplementary if you haven’t watched it yet. We would like to hear more from you if you have further questions or comments.
>
> **W1, W2: Limited analysis of retrieval mechanism. Performance may degrade given irrelevant references. Analyze scenarios where the input concept conflicts with the reference.**
>
> - Regarding the retrieval mechanism, we have detailed our retrieval data and method in Appendix (A.1) and analyzed the limitation of the current retrieval mechanism based on Uni3D in Fig. 15 and Appendix (B). We want to clarify that the specific retrieval mechanism is not our contribution and is replaceable. The main contribution of Phidias is the proposed reference-augmented multi-view diffusion model for enhancing the generative quality, generalization ability, and controllability using an additional 3D reference that can be achieved in many ways, such as (1) retrieved from a 3D dataset, (2) provided by users (Fig. 9), and  (3) created manually (Fig. 10).
>
> - Regarding the performance given irrelevant reference and analysis on scenarios where the input concept conflicts with the reference, please refer to the 2nd point of Common Response above for more details. In short, Phidias is designed to adapt to 3D references with different similarity levels thanks to the proposed meta-ControlNet and dynamic reference routing. If the retrieved references are irrelevant,  phidias’s performance would not degrade but equal existing SOTA image-to-3D generation methods without 3D reference, as shown in Fig. 7 (b). However, retrieved reference models with misaligned poses or structures can lead to undesired outcomes, as the two failure cases shown in Fig. 15. We believe advanced retrieval mechanisms and a larger 3D database would alleviate this problem, which we leave as further works (L944-945).
>
> **Q1: Why does the point cloud for retrieval only contain 10K points?**
>
> We follow the training setting of Uni3D to sample 10K points from the mesh surface with colors to leverage the pre-trained prior in the Uni3D encoder fully.
>
> **Q2: Why are the results of Fig. 14 (currently Fig. 15) different from Fig. 4?**
>
> Fig. 15 shows the actual CCMs of reference we used as a condition of the meta-ControlNet, while Fig. 4 shows the meshes of 3D reference under poses that better present their 3D structures. The actual poses we used to render CCMs of 3D references in Fig. 4 match the input image.
>
> **Q3: Inference speed and resource**
>
> As suggested, we have detailed the inference speed in Appendix (A.3, L858) and the 3rd point of Common Response. In short, Phidias can generate a final 3D model from the input image **in less than 10 seconds**, including 9 seconds for multi-view generation, 0.03 seconds for image-based 3D retrieval, 0.4 seconds for rendering the CCM maps, and 0.02 seconds for sparse-view 3D reconstruction. The inference time is tested using a single NVIDIA A100 GPU. The precomputed embeddings of 3D models in the dataset are stored as a tensor of size Nx1024, where N is the size of the dataset. The retrieval latency will not significantly increase even if we scale up the dataset thanks to the super-fast matrix multiplication using GPU.
>
> Thank you again for your efforts in reading and reviewing our paper. We hope our explanation has alleviated your concerns. Please feel free to reply if you have more questions regarding our paper.
>
> Bests,
>
> Authors

---

> > ### Author Response · Authors · 2024-11-25
> > **Kindly Invitation to Join Rebuttal Discussions**
> >
> > Dear Reviewer eNa9,
> >
> > Thank you once again for your valuable comments on our paper. A few days ago, we submitted a revised version of the paper, highlighting the revisions in red. We also provided a common response for all reviewers and individual responses for each reviewer. Additionally, we have included a demo video in the supplementary materials to showcase our results. We hope our responses address your concerns.
> >
> > As the rebuttal deadline approaches, we kindly invite you to join the rebuttal discussions. Your invaluable feedback and suggestions are greatly appreciated and will help us refine our work further.
> >
> > Best regards,
> >
> > The Authors

---

> > > ### Author Response · Authors · 2024-11-29
> > > **Kindly Invitation to Join Rebuttal Discussions**
> > >
> > > Dear Reviewer eNa9,
> > >
> > > Thank you once again for your valuable comments on our paper. A few days ago, we submitted a revised version of the paper, highlighting the revisions in red. We also provided a common response for all reviewers and individual responses for each reviewer. Additionally, we have included a demo video in the supplementary materials to showcase our results. We hope our responses address your concerns.
> > >
> > > As the rebuttal deadline approaches, we kindly invite you to join the rebuttal discussions. Your invaluable feedback and suggestions are greatly appreciated and will help us refine our work further.
> > >
> > > Best regards,
> > >
> > > The Authors

---

> > > > ### Comment · Reviewer_eNa9 · 2024-12-02
> > > >
> > > > Dear Authors,
> > > >
> > > > Thank you for your detailed response and for addressing my concerns. I appreciate the clarity provided regarding the retrieval mechanism, inference efficiency, and the differences in figures.
> > > >
> > > > While I find your contributions valuable, I believe some aspects, such as the handling of misaligned references and the scalability of the retrieval mechanism, could be explored in greater detail to make the paper even more compelling. Nevertheless, I recognize the novelty and potential impact of your work and will maintain my current score.
> > > >
> > > > Best regards

---

### Official Review · Reviewer_mgWn · 2024-10-25

**Soundness:** 3
**Presentation:** 3
**Contribution:** 3
**Rating:** 5
**Confidence:** 3

**Summary:**

The paper introduces a generative model designed for creating 3D content from text, image, and 3D conditions with reference-augmented diffusion. Phidias aims to enhance the quality, generalization, and controllability of 3D generation by leveraging existing 3D models as references. The model integrates three key components: meta-ControlNet for dynamically adjusting conditioning strength, dynamic reference routing to address misalignments between input images and 3D references, and self-reference augmentations for self-supervised training with a progressive curriculum.

These components collectively improve generative performance over existing methods, offering a unified framework for 3D generation using diverse inputs. Phidias enables various applications, including retrieval-augmented image-to-3D, text-to-3D, theme-aware 3D-to-3D generation, interactive 3D generation with coarse guidance, and high-fidelity 3D completion.

**Strengths:**

1. This paper proposes a dynamic reference routing approach. Recognizing that reference models do not perfectly correspond to the conceptual image, this paper utilize a feature pyramid to address this discrepancy. The motivation is that, it has been widely observed during the reverse diffusion process that the coarse structure of a target image is established in the early, high-noise timesteps, while finer details emerge progressively as the timesteps advance. This motivates to start with low-resolution reference CCMs at high noise levels.
2. This paper facilitates a range of downstream applications, including retrieval-augmented image-to-3D, text-to-3D, theme-aware 3D-to-3D generation, interactive 3D generation with coarse guidance, and high-fidelity 3D completion.

**Weaknesses:**

1. Dependence on Retrieval Quality: As stated by the author The performance of Phidias is contingent upon the quality and relevance of the retrieved 3D reference models, which may not always be optimal.
2. Database Limitations: The current 3D database used for retrieval is limited in size, which could restrict the diversity and accuracy of reference models available for augmentation. Moreover, the model necessitates an additional dataset for inference, which may further restrict its applicability.

**Questions:**

1. In line 186, there is a missing citation regarding the canonical coordinate map. I believe this map is also referred to as the NOCS map, as introduced in reference [a].

[a] Normalized Object Coordinate Space for Category-Level 6D Object Pose and Size Estimation

---

> ### Author Response · Authors · 2024-11-20
> **Reponse to R-mgWn**
>
> Dear Reviewer mgWn,
>
> Thanks for your constructive comments.  We hope our responses below address your concerns about our paper. We kindly invite you to watch the attached demo video in the supplementary if you haven’t watched it yet. We would like to hear more from you if you have further questions or comments.
>
> **W1: Dependence on Retrieval Quality**
>
> As a reference-augmented generation pipeline for image-to-3D generation, the upper bound of Phidias depends on the quality of retrieved 3D reference. However, as stated in the 2nd point of the Common Response (Please refer to the common response above for more details), Phidias's lower bound equals SOTA image-to-3D generation without 3D reference, even if the 3D reference is significantly different from the concept image. The reason is that, during inference, meta-ControlNet will produce adaptive control signals based on the similarity between the 3D reference and concept image (L229-232), and dynamic reference routing will try to alleviate local conflicts (L406-410), which allows Phidias to depend less on retrieval quality for stable inference (Fig. 7 (a) and (b)).
>
> **W2: Database Limitations**
>
> Note that our promising results are based on a small 3D database of 40K objects. This can be significantly enhanced if the dataset is scaled to the whole Objaverse, ObjaverseXL, and other 3D databases. Similar to RAG in a large language model, which requires an additional large-scale database (all information throughout the network) for achieving promising results, we believe that the advantages of incorporating an additional database for RAG-based 3D content generation outweigh its potential disadvantages, given its potential industrial value and scalable ability.
>
> **Q1: Missing citation regarding CCM**
>
> Thank you for pointing this out. We have included the reference [a] you suggested in L183. In our first manuscript, we use the term “canonical coordinate map (CCM)” following CRM [Wang et al., 2024a].
>
> Thank you again for your efforts in reading and reviewing our paper. We hope our explanation has alleviated your concerns. Please feel free to reply if you have more questions regarding our paper.
>
> Bests,
>
> Authors

---

> > ### Author Response · Authors · 2024-11-25
> > **Kindly Invitation to Join Rebuttal Discussions**
> >
> > Dear Reviewer mgWn,
> >
> > Thank you once again for your valuable comments on our paper. A few days ago, we submitted a revised version of the paper, highlighting the revisions in red. We also provided a common response for all reviewers and individual responses for each reviewer. Additionally, we have included a demo video in the supplementary materials to showcase our results. We hope our responses address your concerns.
> >
> > As the rebuttal deadline approaches, we kindly invite you to join the rebuttal discussions. Your invaluable feedback and suggestions are greatly appreciated and will help us refine our work further.
> >
> > Best regards,
> >
> > The Authors

---

> > > ### Author Response · Authors · 2024-11-29
> > > **Kindly Invitation to Join Rebuttal Discussions**
> > >
> > > Dear Reviewer mgWn,
> > >
> > > Thank you once again for your valuable comments on our paper. A few days ago, we submitted a revised version of the paper, highlighting the revisions in red. We also provided a common response for all reviewers and individual responses for each reviewer. Additionally, we have included a demo video in the supplementary materials to showcase our results. We hope our responses address your concerns.
> > >
> > > As the rebuttal deadline approaches, we kindly invite you to join the rebuttal discussions. Your invaluable feedback and suggestions are greatly appreciated and will help us refine our work further.
> > >
> > > Best regards,
> > >
> > > The Authors

---

### Official Review · Reviewer_tCL1 · 2024-11-03

**Soundness:** 3
**Presentation:** 3
**Contribution:** 3
**Rating:** 6
**Confidence:** 4

**Summary:**

This paper presents a new 3D generation approach that utilizes a reference 3D model to enhance the generation process. The method comprises three key components: (1) Meta-ControlNet (2) Dynamic Reference Routing (3) Self-Reference Augmentation. Experimental results demonstrate the effectiveness of this method in enhancing 3D generation quality, generalization ability, and controllability.

**Strengths:**

- Gathering information from a 3D reference model is straightforward.
- The evaluation of generation quality shows the effectiveness of the approach.
- The paper is well-written and easy to follow.

**Weaknesses:**

- In the evaluation, only generation quality is assessed, with no evaluation of generation speed. For Phidias, retrieving information from the reference 3D model requires both 3D model retrieval and rendering of the canonical coordinate maps, which introduces additional latency. However, the paper does not report specific latency overheads.
- From Table 3, the effectiveness of dynamic routing appears to be limited.

**Questions:**

- With the dynamic reference routing mechanism, the meta-ControlNet should be capable of handling inputs at various resolutions. Is the meta-ControlNet trained on images of different resolutions, or is it trained solely on the highest resolution and expected to generalize to lower resolutions?

---

> ### Author Response · Authors · 2024-11-20
> **Response to R-tCL1**
>
> Dear Reviewer tCL1,
>
> Thank you for your constructive suggestions. We hope our responses below address your concerns about our paper. We kindly invite you to watch the attached demo video in the supplementary if you haven’t watched it yet. We would like to hear more from you if you have further questions or comments.
>
> **W1: Inference speed**
>
> As suggested, we have detailed the inference speed in Appendix (A.3, L858) and the 3rd point of Common Response. In short, retrieving the 3D model and rendering the CCMs do not introduce significant latency during inference. Phidias can generate a final 3D model from the input image **in less than 10 seconds**, including 9 seconds for multi-view generation, 0.03 seconds for image-based 3D retrieval, 0.4 seconds for rendering the CCM maps, and 0.02 seconds for sparse-view 3D reconstruction. The inference time is tested using a single NVIDIA A100 GPU. The retrieval process is fast as we only need to compare the query vector of an input image (dim=1x1024) with precomputed embeddings of 3D models in the dataset (dim=Nx1024, where N is the size of the dataset) using cosine similarity, as stated in L783-785. The retrieval latency will not significantly increase even if we scale up the dataset thanks to the super-fast matrix multiplication using GPU.
>
> **W2: The effectiveness of dynamic routing in Tab. 3 seems limited**
>
> Dynamic reference routing focuses on dealing with local conflicts (L235-239), which may not bring significant performance enhancement when computing quantitative metrics but is very important for preserving local details of input concept images during inference. As shown in Fig. 6  (b), the wire in the astronaut will be ignored without dynamic reference routing because the 3D reference does not have this wire.
>
> **Q1:  Is the meta-ControlNet trained on images of different resolutions?**
>
> Yes, meta-ControlNet is trained on images of different resolutions depending on the sampled timestep (noise levels) during each training step. We have detailed meta-ControlNet training with dynamic reference routing in Appendix (A.2, L804-808).
>
>
> Thank you again for your efforts in reading and reviewing our paper. We hope our explanation has alleviated your concerns. Please feel free to reply if you have more questions regarding our paper.
>
> Bests,
>
> Authors

---

> > ### Comment · Reviewer_tCL1 · 2024-11-24
> >
> > Thank you for your response! I am glad to see the 3D retrieval takes only 0.03s. I noticed that the LGM paper reports a Gaussian generation time of approximately 1 second. Considering that your approach increases both the number of input images and the resolution, I am curious how you achieve sparse-view 3D reconstruction in just 0.02 seconds.

---

> ### Author Response · Authors · 2024-11-24
>
> Hi Reviewer tCL1,
>
> Thank you for your response! In our experiment, 0.02s is an approximate time for generating 3D Gaussians from sparse-view images. **The time reported in the LGM paper includes additional time for saving the Gaussians into a .ply file and rendering a visualization video.**
>
> For a fair comparison, we report the step-wise time for Gaussian generation, saving, and rendering given the mushroom teapot in Fig. 7 (b) of our revised paper, using the same device (a single A100 GPU).
>
> The tested time of **LGM** is 0.019s (Gaussian generation), 0.095s (saving .ply), and 0.529s (render a video with 180 frames).
>
> The tested time of **Phidias** is 0.023s (Gaussian generation), 0.143s (saving .ply), and 0.756s (render a video with 180 frames).
>
> As can be seen, after increasing the number of input images and the resolution, the time for sparse-view 3D reconstruction has increased but is still fast.
>
> Thank you again for your response! We hope our additional experiments and discussions can address your concerns regarding the inference speed of Phidias. Please feel free to let us know if you have any other questions.
>
> Best regards,
>
> Authors

---

> > ### Comment · Reviewer_tCL1 · 2024-11-24
> >
> > Thanks! All my concerns are solved. I have increased my rating.

---

### Official Review · Reviewer_phbC · 2024-11-04

**Soundness:** 3
**Presentation:** 3
**Contribution:** 3
**Rating:** 6
**Confidence:** 4

**Summary:**

The paper present a reference-augmented method using retrieved 3D objects as auxiliary guidance to reduce the ill-posed nature of the single-view 3D reconstruction task. The method leverages multi-view diffusion zero123++ for the generation of 6 novel view images, and scale-up the LGM for 3D reconstruction. The main contribution of the paper is to use input concept image (typically lack of 3D information of the back or side view, etc.) to refer a similar 3D objects from their 3D database (40K objects selected from Objaverse), the 6 canonical coordinate maps (CCM) from referred 3D object is used to generate better multi-view images. Meta-ControlNet with dynamic resolution of the intake CCM during diffusion sampling contribute to the misalignment dilemma between the concept images and referred 3D object. The results show that the method outperform other SOTA models in image-to-3D tasks given the proper 3D object reference. And it may extend to text-to-3D, and 3D-to-3D applications.

**Strengths:**

1. The framework extends the controllability of current image-to-3D models. The model retrieves an auxiliary 3D object as guidance to improve the 3D reconstruction quality. The results shows outperformance than current SOTA models.

2. The paper propose meta-controlnet, and self-reference inference to let model learn how to resolve the conflict between retrieved objects and concept front-view image. Given the setting by the paper claimed, the model is able to provide reasonable 3D generation results matches on both sides.

3. The model is able to extend to text-to-3D and 3D-to-3D generation tasks, which may contribute to more broad applications.

**Weaknesses:**

1. (model behavior) The model is trained with self-augmented ground-truth 3D objects, in order to handle the misalignment dilemma. However, as the A.4 specified, the augmented ones are still very similar to the ground truth. Moreover, all showcases are with no explicit conflicts between concept image and 3D objects. This raises the question of how a bad retrieval influence the model performance, which is actually common in practical case (as line 878-879). More comprehensive analysis or comparisons are necessary.

2. (retrieval robustness) The quality of the retrieval directly influence the improving of the generation quality, especially on the back or side views. I elaborate in detail in Question-1. If the user have either a out-of-retrieval-database front-view concept image, or the retrieval brings an unreasonable objects, the model output may not be as good as claimed. As Fig.7 shows, random reference as an example of non-similar retrieval may provide unreasonable results. This restricts the usage of the method.

3. (restricted/unfair experiment setting) Followed by weakness-1, the paper constrains the input image as only-front-view (line 221-222) to all competitor models. A random view, or 2-4 views as other related works defined, will contain more geometric information of the 3D details of the object. If not restricting to only-front-view input, using other models which do not require an auxiliary 3D object input like OpenLRM may already provide reasonable results. Their results may be better than the results with no reference in Fig.1, and Fig.7, and SOTA results in Fig. 5.

4. (restricted/unfair experiment setting) In the model evaluation section, all showcase examples meet the optimal situation of the framework. (a) The retrieved 3D objects is very similar to the concept image in the front view, (b) there is no/little conflicts on the other views. Given that, the improvement of the quantitative results in Tab.1 is still marginal. The row "Ours (GT Ref.)" may not be a reasonable comparison, since it provides ground truth geometric information in 6-view. I am not surprise it provides better generation results from the unseen view.

5. (model usage) When there is considerable conflict between retrieved 3D objects and concept images (as line 878-879, it is actually very common), which side would the model follow the most? Potential users may want to see more examples of practical cases. And from a point of view as graphics designer, it will be better to provide a controlling nob for user to decide the extent of guidance from either side, rather let the model itself decides the extent. Right now, it is still uncontrollable.

6. (novelty) The novelty of the paper concentrated on the meta-controlnet and reference-based inference. However, given the weakness aforementioned, it may solve an restricted task or with limited contribution to the real problem, and compare to SOTAs in an unfair setting. Moreover, the reason to use diffusion models, and following reconstruction model may need more elaborations.

**Questions:**

Great work! I elaborate more based on weakness, and provide some suggestions and questions.

1. What is the main research goal of the paper? Is it to improve the quality of the 3D reconstruction on un-seen views (which indicates the output model should faithfully obey the input concept image), or it provides more customization freedom by given auxiliary 3D object reference (which indicates the output should lean to the given 3D objects.). Based on a full reviewing of the paper, my understand is the paper lean to the former one (correct me if I am wrong). So when there is conflict between retrieved 3D objects and concept images, which side would the model follow the most? If I am a graphics designer, a useful one will be a model doing either way, rather than a model-decided interpolation between two sides.

2. Is it a correct understand that the meta-controlnet is actually to learn a 3D texture mapping, which utilize the pixel-based texture in the concept image, map it to the retrieved 3D object surface based on 6-view CCMs? In that, the best way to phrase the contribution may like "using the front-view image as 2D texture guidance to generate unseen views given the 6-view CCM priors".

3. Follow Question-2, would you elaborate more about the role and reason to use diffusion model here? My understanding is, because there is conflict if we directly use 3D texture mapping using concept images and 6-view CCMs, so you let the diffusion process handle those conflicts. Any reason to exclude other pixel-based networks or traditional CG methods?

4. There are a lot of necessary details in the supplementary, like details of retrieving, dataset, etc. It will be better to move them into the main paper, even some of details may raise questions from reviewers. There is still a half-page blank now.

 I reserve the change of grades upon authors' responses to my reviewing.

---

> ### Author Response · Authors · 2024-11-20
> **Response to R-phbC**
>
> Dear Reviewer phbC,
>
> Thanks for your careful review and constructive comments. We appreciate you recognize our paper as a great work that provides additional controllability for current image-to-3D generation models. We note that you still have several concerns about the performance of Phidias, the majority of which come from misunderstandings regarding the methods and experimental settings. We would like to provide more details regarding your questions and hope your concerns can be addressed.
>
> **W1: Model behavior**
>
> You said, “As A.4 (current A.5 after we updated the pdf) stated, the augmented ones are still very similar to the ground truth.” Yes, but the self-augmented 3D objects are only part of our **curriculum training strategy** (L261-265). As stated in Sec. 3.4, during training, we start from these self-augmented 3D objects to avoid the ‘idleness’ of reference, i.e., pushing our model to learn to use the reference frist because they are similar to the ground truth. Then, we will increase augmentation strength and incorporate retrieved references (L879-880) to simulate significant misalignment during inference. Thus, Phidias does see references with different similarity levels during training. Once trained, Phidias performs well with a variety of references, even those retrieved ones that are not very similar (L265).
>
>
>  **W2: Retrieval robustness**
> As suggested, we have added a more comprehensive analysis of the bad influence of the unreasonable retried reference with explicit conflicts with the confect image, as shown in Fig. 7 (b). Please refer to the second point of our Common Response for details. Here, we want to emphasize that the lower bound of our performance equals an image–to-3D generation method without reference since our model is designed to ignore the bad influence of unreasonable references. That is also why we achieve promising results with a small retrieval database. If we scale up the size of the database, our model will benefit more from the retrieved reference as the similarity levels increase.
>
> **W3,W4: Restricted/unfair experiment setting**
>
> - Regarding the input view, Phidias can improve the results significantly given front-view input image, but it also works well with random-view inputs, as shown in Fig. 14 (camel) and Fig. 18 (robot, dinosaur, pig, horse, chest, car, etc.). As stated above, the lower bound of Phidias equals to SOTA image-to-3D generation methods without reference. If not restricted to only-front-view input (either a random view or 2-4 views), other methods will not be better than Phidias, i.e., if their results are reasonable, the results of Phidias will also be reasonable or even better. The core of Phidias’s design is to use an additional 3D reference to improve the results’ quality and controllability **adaptively**, which is unique from all previous methods.
>
> - Regarding your concern about the quantitative results in Tab. 1, there should be a misunderstanding. (a) the retrieved 3D objects, in most cases, are not very similar to the concept image in the front view when we compute the metrics; (b) the differences between the retrieved reference and GT will result in a marginal reconstruction loss, even the results of Phidias are plausible and better than the baselines. Thus, we do not aim to surprise you with better results for the row “Ours (GT)” and we also know that the row “Ours (Retrieved Ref.)” is still marginal. What we want to emphasize in Tab. 1 is that **the actual performance of Phidias should be between “Ours (Retrieved Ref.)”  and “Ours (GT Ref.)”.** We have included these discussions in the revised paper (L358-361). We have also conducted additional user studies to evaluate our method based on human preference (Tab. 2 and L362-366).
>
> **W5: Model usage**
> When there are conflicts between retrieved 3D objects and concept images, Phidias is trained to adaptively adjust the guidance strength of the 3D reference according to their similarity levels (L229-232). If there are significant conflicts, Phidias will follow the concept image, as shown in Fig.7 (b), because the default setting of our task is image-to-3D generation, and we need to follow the image first. However, Phidias also allows users to adjust the guidance strength manually, just like the “control knob” as you said. We have included this functionality as a new application in Fig. 12 and L483-485.
>
> **W6: Novelty**
> We hope our explanation above and the 1st point of Common Response have alleviated your concerns regarding our novelty. The novelty of our paper is to bring the concept of RAG into 3D generation. Meta-ControlNet and reference-based inference are solutions for incorporating conflicted 3D reference into the image-to-3D generation pipeline, i.e., the Misalignment Dilemma in L78-79. **Regarding why we use diffusion models and the following reconstruction model**, please refer to our Q2 and Q3 responses below.

---

> > ### Author Response · Authors · 2024-11-20
> > **Response to R-phbC (part-2)**
> >
> > **Q1: What is the main research goal of the paper?**
> > Our main objective is reference-augmented image-to-3D generation, which can be further extended to text-to-3D and 3D-to-3D. Thus, the output model should (1) faithfully obey the input concept image, and (2) the output should lean to the given 3D objects **in unseen views**, which indicates that the output should learn to adaptively utilize the 3D reference to provide more controllability, improve generation quality, and enhance generalization ability **in unseen views**. If significant conflicts exist between 3D reference and concept image, the default setting is to follow the concept image most. However, Phidias can also do either way by providing a “control knob”, as we mentioned in W5 above (see Fig. 12 and L483-485).
> >
> > **Q2, Q3: What does meta-ControlNet do? Why use the diffusion model?**
> >
> > You may have misunderstood our task and the model design. As mentioned above, our task is to generate a 3D model (both geometry and texture) based on a single image instead of texturing an existing 3D mesh. The 3D reference is used to enhance the performance and provide additional controllability in unseen views.
> >
> > We use the diffusion model because we designed our model following the widely used pipeline for image-to-3D generation (e.g., InstantMesh, LGM, CRM), which combines multi-view diffusion and sparse-view 3D reconstruction. This pipeline achieves state-of-the-art performance in image-to-3D generation with the help of 2D diffusion priors and transformer/U-Net-based large 3D reconstruction models.
> >
> > Since the multi-view diffusion model decides the final geometry and appearance of the generated 3D object, we incorporate the reference guidance into it via a ControlNet architecture conditioned on multi-view CCMs. However, the vanilla ControlNet is designed for image-to-image translation that demands the generated images to closely follow the conditions (in our task are CCMs), which is not aligned with our task because we treat the reference model as auxiliary guidance without requiring precise alignment with the reference model (**We have explicitly explained this point in Introduction L83-L86**). Besides, during inference, the retrieved 3D objects may differ from the concept image to various extents, so we design meta-ControlNet upon the vanilla ControlNet to produce adaptive control signals based on the similarity between the concept image and 3D reference (L87-88).
> >
> > **Q4: Move more details from the Appendix to the main paper**
> >
> > We kept our main paper to 9 pages with some blank as suggested by ICLR (https://iclr.cc/Conferences/2025/CallForPapers) because we need to keep some additional space for author information and acknowledgments in the final camera-ready version. We are not trying to avoid potential reviewer questions by moving some important details into the Appendix. If we have space in the final camera-ready version, we would like to move more details into the main paper.
> >
> > Thank you again for your efforts in reading and reviewing our paper. We hope our explanation has alleviated your concerns. Please feel free to reply if you have more questions regarding our paper.
> >
> > Bests,
> >
> > Authors

---

> > > ### Author Response · Authors · 2024-11-25
> > > **Kindly Invitation to Join Rebuttal Discussions**
> > >
> > > Dear Reviewer phbC,
> > >
> > > Thank you once again for your valuable comments on our paper. A few days ago, we submitted a revised version of the paper, highlighting the revisions in red. We also provided a common response for all reviewers and individual responses for each reviewer. Additionally, we have included a demo video in the supplementary materials to showcase our results. We hope our responses address your concerns.
> > >
> > > As the rebuttal deadline approaches, we kindly invite you to join the rebuttal discussions. Your invaluable feedback and suggestions are greatly appreciated and will help us refine our work further.
> > >
> > > Best regards,
> > >
> > > The Authors

---

> > > > ### Comment · Reviewer_phbC · 2024-11-25
> > > > **Response to Authors**
> > > >
> > > > Thanks for your responses of my questions. Most of my concerns have been addressed. I will keep my scores for the current version. Thanks for your efforts and active responses. Good luck!

---

### Author Response · Authors · 2024-11-20
**Common Response**

We would like to thank all reviewers for their valuable comments. We appreciate that reviewers recognize our ideas for RAG-based 3D generation as novel, with good results and various downstream applications. **We have revised our paper based on the reviewers’ comments in red (see the updated pdf).** Before we proceed with a detailed, point-by-point response for each reviewer, we hope that the reviewers and the Area Chair could browse through this blog content first, which primarily addresses potential misunderstandings and similar concerns about our paper. For other individual questions, we will answer one-by-one under each reviewer’s section.

**1. Regarding the potential impact of our paper**

As the first work that brings the concept of retrieval-augmented generation (RAG) into the field of 3D generation, we achieve significant improvements in generative quality, generalization ability, and controllability. We kindly invite the reviewers and the Area Chair to watch the attached demo video in the supplementary if you haven’t watched it yet. Note that these promising results are achieved using a small retrieval database of only 40K 3D objects, which could be significantly enhanced if we further scale up this database (L304). We believe our paper has industrial values and could greatly inform future work in RAG-based 3D/4D and video generation (as stated by Reviewer-Ef4X).


**2. Common Question 1: Generation robustness given dissimilar 3D reference with explicit conflicts (R-phbC, R-mgWn, R-eNa9, R-Ef4X)**

We have carefully designed the meta-ControlNet and dynamic reference routing for robust inference, given that the 3D references have different similarities. While dynamic reference routing focuses on dealing with local conflicts (L406-410), Meta-ControlNet is designed to produce adaptive control signals so that Phidias can globally adjust the reference strength given 3D reference with different similarity levels (L229-232). Theoretically, with these two designs, Phidias can be robust to references that are significantly different from the concept image. We have also analyzed the influence of 3D reference with varying levels of similarity in Fig. 7 (a) and Sec. 4.2. Phidias can still generate plausible results even with a random reference (L422-424).

Some reviewers suggest including examples using references with more explicit conflicts. First, we have already included such cases in the Appendix, such as the gray robot (1st row, 1st col) and ceramic incense vessel (2nd row, 1st col) in Fig. 17 and the wooden pig (2nd row, 2nd col) and multi-feet robot (4th row, 2nd col) in Fig. 18. Second, as suggested, we further add additional results with more explicit conflicts in Fig. 7 (b) and discussions in L424-426 to demonstrate the robust performance of our method further. As can be seen, **the lower-bound performance of Phidias, when the reference is significantly different from the concept image, equals SOTA image-to-3D without reference**. Note that, in most cases, the retrieved 3D model will not be totally different from the image by retrieval. The performance would be more stable and significantly enhanced if the dataset were scaled to the whole Objaverse, ObjaverseXL, and other 3D databases.

**3. Common Question 2: Inference speed (R-tCL1, R-eNa9)**

As suggested, we detail the inference speed in Appendix (A.3, L858). In short, similar to existing two-stage image-to-3D generation methods, Phidias can generate a final 3D model from the input image **in less than 10 seconds**, including 9 seconds for multi-view generation, 0.03 seconds for image-based 3D retrieval, 0.4 seconds for rendering the CCM maps, and 0.02 seconds for sparse-view 3D reconstruction. The inference time is tested using a single NVIDIA A100 GPU. The retrieval process is fast as we only need to compare the query vector of an input image (dim=1x1024) with precomputed embeddings of 3D models in the dataset (dim=Nx1024, where N is the size of the dataset) using cosine similarity, as stated in L783-785. The retrieval latency will not significantly increase even if we scale up the dataset thanks to the super-fast matrix multiplication using GPU.

**4. Summary of changes**

Please review the updated paper to see if your concerns are addressed. Notable changes include:
- We have included additional results in Fig. 7 (b) and discussions in L424-426 to demonstrate the robustness of Phidias given reference with explicit conflicts.
- We have added a new application in L483-485 and Fig. 12 to provide a “control knob” for users to decide the reference strength manually, as suggested by R-phbC.
- We have included the details of inference speed in A.3, L858.
- We have included additional discussions to further explain the results of Tab. 1 in L358-361.
- We have included the reference of “Normalized Object Coordinate Space for Category-Level 6D Object Pose and Size Estimation” in L183, as suggested by R-mgWn.


Best regards,

Authors

---

### Author Response · Authors · 2024-12-02
**Invitation to Join Rebuttal Discussions (Last Day)**

Dear Reviewer eNa9 and Reviewer mgWn,

I hope this message finds you well. Thank you once again for your valuable comments on our paper. **As the last day for reviewer responses is tomorrow, we kindly invite you to join the rebuttal discussions today.**

We have submitted a revised version of the paper, along with a common response for all reviewers, individual responses for each reviewer, and a demo video in the supplementary materials. We hope these responses and additional materials address your previous concerns.

We would greatly appreciate your engagement, feedback, and suggestions, which will help us refine our work further. If your concerns have already been addressed, please also consider increasing your ratings.

Thank you very much for your time and consideration.

Best regards,

The Authors

---

### Meta-Review · Area_Chair_6SRt · 2024-12-20

**Metareview:**

This paper proposes a unified approach for text or image guided 3D generation via multi-view diffusion models, which is further guided with a retrieved 3D object, for the very first time. In order to resolve the fundamental conflict caused by the incongruence between the guiding concept image and the 3D object, the authors propose three strategies to mitigate it: a meta-ControlNet to adaptively control the guidance from the 3D model's CCM images; using multiple resolutions for the 3D object as the reverse diffusion process proceeds and using a perturbation-based strategy for training with progressively more misaligned 3D objects in a self-reconstruction training setting. The work shows state-of-the-art performance and many novel downstream applications.

The significance of this work includes that it is the first unified framework to include a guiding 3D object for 3D generation, which can be helpful in many downstream application such as asset creation by artists in interactive settings. It is also broadly applicable to all 3D generation problems.

**Additional Comments On Reviewer Discussion:**

Five reviewers provided the final scores of 6, 6, 5, 6, 8. Their main concerns were around how the quality of the retrieved 3D model affects the quality of the generated results, the speed of the method and its dependence on a 3D dataset. The authors provided extensive discussions and results during the rebuttal phase, which addressed a large majority of the reviewers' concerns. 4/5 reviewers favored accepting this work and appreciated it for its novelty and high-quality results.

The AC concurs and recommends acceptance.

---

### Decision · Program_Chairs · 2025-01-22

Accept (Poster)